

# Distinctive dust weather intensities in North China resulted from two types of atmospheric circulation anomalies

Qianyi Huo[1], Zhicong Yin [123], Xiaoqing Ma[1], Huijun Wang[123]

[1]Key Laboratory of Meteorological Disaster, Ministry of Education / Joint International Research Laboratory of Climate and Environment Change (ILCEC) / Collaborative Innovation Center on Forecast and Evaluation of Meteorological Disasters (CIC-FEMD), Nanjing University of Information Science & Technology, Nanjing 210044, China
[2] Southern Marine Science and Engineering Guangdong Laboratory (Zhuhai), Zhuhai, China
[3]Nansen-Zhu International Research Centre, Institute of Atmospheric Physics, Chinese Academy of Sciences, Beijing, China

*Correspondence to*: Zhicong Yin (yinzhc@nuist.edu.cn)

**Abstract.** Dust weather in North China (NC) has worsened in recent years, posing adverse impacts on the environment, human health, and the economy. In 2021, the "3.15" super dust storm raised Beijing's $PM_{10}$ (particulate matter with a diameter less than 10 μm) concentrations above 7000 μg m$^{-3}$, while 2023 witnessed the highest spring dust weather frequency in nearly a decade. Although previous studies have proposed that synoptic systems such as the Mongolian cyclone and cold high can induce dust weather in NC, there has been less focus on the cold high. Furthermore, the differences in $PM_{10}$ concentrations in NC caused by the two synoptic systems have not been quantified. This study demonstrates that the Mongolian cyclone was responsible for 62.4% of the dust weather in NC, while the remaining 37.6% was primarily caused by the cold high. The dust intensity induced by the Mongolian cyclone was stronger than that of the cold high, with average maximum $PM_{10}$ concentrations of 3076 μg m$^{-3}$ and 2391 μg m$^{-3}$, respectively. The three-dimensional structure of atmospheric circulation anomalies and related dynamic mechanisms of the two types were concluded. A common predictor of the two dust weather types has also been identified. These findings contribute to enhancing the comprehension of dust weather in North China and offer valuable insights for both dust weather forecasting and climate prediction.

## 1 Introduction

Dust weather frequently occurs in spring over North China (NC) and has a number of negative effects on the environment, human health and economy. The strong wind which induces the dust weather, has the potential to inflict severe damage on infrastructure and results in soil erosion, thereby exerting detrimental effects on agricultural productivity (Ahmadzai et al., 2023). Simultaneously, the entrainment of dust particles by the strong wind leads to an increase of $PM_{10}$ (particulate matter with a diameter less than 10 μm) concentration (Krasnov et al., 2016), which reduces air quality and poses a threat to human health, increasing the risk of respiratory and cardiovascular diseases (Lwin et al., 2023). Besides, dust aerosols reduce visibility (Gui et al., 2022), impeding people's travel and adversely affecting traffic safety. In response to global warming, extreme weather has been intensifying and occurring more frequently in China (Yin et al., 2023b). Severe dust storm reoccurred in the





spring of 2021 after an absence for more than 10 years in NC (Zhang et al., 2022). During 14–16 March 2021, the $PM_{10}$ concentration exceeded the monitoring threshold in Ordos (>9985 µg m$^{-3}$) and reached extraordinarily high value in Beijing (>6400 µg m$^{-3}$; Filonchyk, 2021). In the spring of 2023, there was a notable increase in the frequency of dust weather, reaching the highest level observed in recent decades (Yin et al., 2023a). The severe dust storm event during 19–24 March 2023

impacted a large range of area, covering over 4.8 million km$^2$ (Yin et al., 2023a). In general, the dust weather in NC during 2021 and 2023 exhibited characteristics of high intensity, frequent occurrence, and extensive impact.

The generation of dust weather requires both dust source conditions and dynamic conditions. The primary dust sources of dust weather include deserts and sandy areas in arid and semi-arid regions (Huang et al., 2014). In East Asia, the Gobi Desert located in the Mongolian Plateau and northern China is a major source of sandstorms (Zhang et al., 2023). In recent

years, China has made significant progress in combating land desertification (Zhang and Huisingh, 2018). However, during the dust weather events in March and April 2023, over 42% of the dust concentration in NC originated from cross-border transport from Mongolia (Chen et al., 2023a). Strong winds and thermally unstable atmospheric stratification were the primary dynamic conditions for the formation of dust weather (Wu et al., 2023), which facilitated the entrainment and uplift of dust particles (Zhao et al., 2022). The intensity of turbulence and the structure of the boundary layer affect the lifting and diffusion

capabilities of sand and dust (Shao, 2008). The combined effect of thermal (i.e., unstable stratification) and dynamic (i.e., near-surface wind shear) factors enhances turbulent motion, leading to increased wind erosion, which favors the lifting of sand and dust (Wiggs, 2011).

The long-distance transport of dust aerosols in the lower atmosphere is regulated by regional synoptic systems (Huang et al., 2014). Meanwhile, the frequent dust weather activities in East Asia during spring are closely related to mid-latitude

synoptic-scale cyclone activities (Qian et al., 2002). The Mongolian cyclone is the primary synoptic system causing dust weather in NC (Li et al., 2022). Both the severe dust storm events on 15 March 2021 and on 22 March 2023 were attributed to the Mongolian cyclone (Mu et al., 2023; Yin et al., 2023a). The appearance of the Mongolian cyclone triggered strong gusts and unstable thermal conditions, disturbed loose surfaces, and lifted surface dust into the air through upward motion (Tian et al., 2023). As the Mongolian cyclone moved eastward, the northerly winds behind it transported dust from the source area

southward, thereby affecting NC (Takemi and Seino, 2005).

Recent weather and climate studies on the atmospheric circulation systems related to dust weather in NC have primarily focused on the frequency, intensity, and physical processes of the Mongolian cyclone (Wu et al., 2016; Bueh et al., 2022; Chen et al., 2023b; Gao et al., 2024). However, the Mongolian cyclone alone cannot explain all instances of dust weather. A dust weather event in NC on 14 March 2023, caused by a cold high (Fig. S1a), resulted in relatively low $PM_{10}$ concentrations at

1247 µg m$^{-3}$ (Fig. S2). This dust weather event went largely unnoticed. Subsequently, on 22 March, a severe dust storm was brought by the Mongolian cyclone (Fig. S1b). This dust weather led to higher $PM_{10}$ concentrations at 9993 µg m$^{-3}$ (Fig. S2), which garnered more attention (Yin et al., 2023a). Liu et al. (2004) and Yun et al. (2013) subjectively classified dust storms in NC based on atmospheric circulation, while Yi et al. (2021) used the K-means clustering method to classify circulation





patterns of dust weather in the entire northern region of China. They all found that besides the Mongolian cyclone, the cold

high and associated cold front also played significant roles in causing dust weather in NC.

What are the differences in $PM_{10}$ concentration and frequency of dust weather caused by the Mongolian cyclone and cold high? What atmospheric circulation and dynamic mechanisms are responsible for the two types of dust weather? Is there any predictor that can forecast dust weather in NC? This study aims to address the above questions, not only to enhance understanding of the NC dust weather but also to provide references for dust weather forecasting and climate prediction.

## 2 Data and method

### 2.1 Data

The fifth generation European Center for Medium Range Weather Forecasts (ERA5) provided hourly reanalysis meteorology data with the horizontal resolution of 0.25°×0.25° on pressure and surface levels in spring (March, April and May) from 2015 to 2023 (Hersbach et al., 2023). Data at 8:00, 12:00, 16:00, 20:00 (Beijing local time) are selected to calculate the

daily mean values. The variables include geopotential height at 500 hPa (Z500), zonal and meridional winds at 10 m (UV10), zonal and meridional winds at 850 hPa (UV850), zonal winds at 200 hPa (U200), sea level pressure (SLP), temperature at 1000 hPa (T1000), temperature at 850 hPa (T850), vertical velocity at 500 hPa (ω500), 10 m wind gust (Gust10), surface air temperature (SAT), planetary boundary layer height (PBLH), and vertical velocity (ω), zonal and meridional winds, divergence (div), specific humidity (q) from 1000 hPa to 200 hPa on 23 pressure levels.

Hourly observed station $PM_{10}$ concentrations in March, April and May from 2015 to 2023 are derived from China National Environmental Monitoring Centre and publicly accessible at https://quotsoft.net/air/. The Normalized Difference Vegetation Index (NDVI) quantifies the vegetation by measuring the difference between near-infrared and red light. Gridded NDVI data, with a horizontal resolution of 1°×1° in March 2023 were obtained from the National Oceanic and Atmospheric Administration's (NOAA) National Centers for Environmental Information (Vermote, 2019). The Copernicus Climate Change

Service (C3S, 2018) produced seasonal forecast products from European Centre for Medium-Range Weather Forecasts (ECMWF) SEAS5.1, comprising a total of 25 ensemble members. In this study, the Z500 and SLP data for spring were initialized annually on February 1st (one-month lead) with a temporal resolution of 12 hours and a spatial resolution of 1°×1°. The daily ensemble mean data for each variable is used in this study.

### 2.2 Method

According to the synoptic definition of the extratropical cyclone (Shou, 2006), a Mongolian cyclone is identified by a central sea level pressure within the region of 40–55°N, 100–125°E that does not exceed 1010 hPa and is lower than the sea level pressure at the surrounding eight grid points. Furthermore, the average pressure gradient in a 5×5 grid centered around the cyclone should be equal to or greater than 0.55 hPa per 100 km. The vertical air temperature difference (VATD) was defined as T1000–T850, which indicated the thermal atmospheric instability. The vertical transport of westerly momentum





was defined as $\partial(u\omega)/\partial P$, $\partial(u\omega)/\partial P < 0$ represents downward fluxes (Zhong et al., 2019). The composites of the variables were computed based on the daily mean values. All anomalies were calculated relative to the daily mean values in spring from 2015 to 2023. The correlation coefficients in this study were calculated using Pearson correlation.

## 3 PM$_{10}$ concentrations related to regional synoptic systems

The maximum PM$_{10}$ concentration serves as an indicator of dust weather intensity, reflecting the greatest impact of dust
weather on NC (34–42°N, 105–120°E). The changes in daily maximum PM$_{10}$ concentrations observed in NC during spring 2015–2023 are illustrated in Figure S2. The first and third quartiles of the PM$_{10}$ concentration series during spring 2015–2023 were 426.5 μg m$^{-3}$ and 1019 μg m$^{-3}$, respectively. Referring to selected percentile values, two PM$_{10}$ concentration thresholds were chosen at 500 μg m$^{-3}$ and 1000 μg m$^{-3}$. Periods when PM$_{10}$ concentrations exceeded 1000 μg m$^{-3}$ were defined as high concentration periods, while periods below 500 μg m$^{-3}$ were categorized as low concentration periods. It can be observed that
the spring PM$_{10}$ concentrations exhibited distinct periods of high and low concentrations. The high concentration periods had relatively short durations (i.e., average of 1.97 days), while the low concentration periods had relatively longer durations (i.e., average of 2.39 days). During high PM$_{10}$ concentration periods, there were sharp increases in PM$_{10}$ concentrations. In this study, the dates corresponding to the maximum PM$_{10}$ concentrations during high PM$_{10}$ concentration periods were defined as Dust days (Fig. S2). Conversely, the dates corresponding to the minimum PM$_{10}$ concentrations during low PM$_{10}$ concentration
periods were defined as Non-Dust days (Fig. S2).

The primary surface synoptic system leading to dust weather in NC was the Mongolian cyclone (Li et al., 2022). Additionally, cold high systems along with accompanying cold fronts could also contribute to dust weather affecting NC (Yun et al., 2013). By objectively identifying the presence of the Mongolian cyclone, Dust days can be further classified into two categories. As depicted in the composite SLP fields for the two types, the main surface synoptic systems for the two types of
Dust days were the Mongolian cyclone and cold high (Fig. S1c, d). Therefore, the two types of Dust days were respectively named Mongolian Cyclone (MC) type and Cold High (CH) type. From 2015 to 2023, Dust days in April and May were more frequent compared to March (Fig. 1a). The highest number of Dust days occurred in May, followed by April, with March having the fewest occurrences (Fig. 1a). The MC type was the predominant weather pattern causing dust weather in NC during 2015–2023, accounting for 62.4% of the total days in spring (Fig. 1a). In May, the proportion of MC days was the highest at
71.7%, followed by April at 57.9%, and March at the lowest with 53.8% (Fig. 1a). Figure 1c illustrates the temporal distribution of MC days, CH days, and Non-Dust days in spring from 2015 to 2023. It was evident that the two types of Dust days exhibited discontinuous characteristics in time each year.

Both the MC type and CH type exhibited high PM$_{10}$ concentrations in NC (Fig. S3c, d). Compared to the CH type, the MC type resulted in higher PM$_{10}$ concentrations and showed more pronounced extremes (Fig. 1b). The outliers in the PM$_{10}$
concentration boxplot in Fig. 1b included the severe dust storms on March 15 2021, and March 22 2023, reflecting the extremity of these two dust events. During Dust days, the dust particles that caused the increase in PM$_{10}$ concentrations



originated from the dust source area. From the NDVI, it can be observed that there was a significant lack of vegetation cover in the northwest direction outside NC (Fig. S3a). This area (NDVI < 0.1), serving as a dust source region, could provide favorable conditions for dust emissions (Wang et al., 2021). From the spatial distribution differences in $PM_{10}$ concentrations,

it can be observed that the MC type resulted in relatively higher $PM_{10}$ concentrations in the northern part of NC, especially near the vicinity of the dust source area (Fig. 1b). Additionally, the Northeast China also experienced significant impacts during MC days (Fig. 1b). In contrast, the CH type led to higher $PM_{10}$ concentrations in the southern part of NC and further southward areas, with the influence of dust particles leaning more towards the south compared to MC days (Fig. 1b).

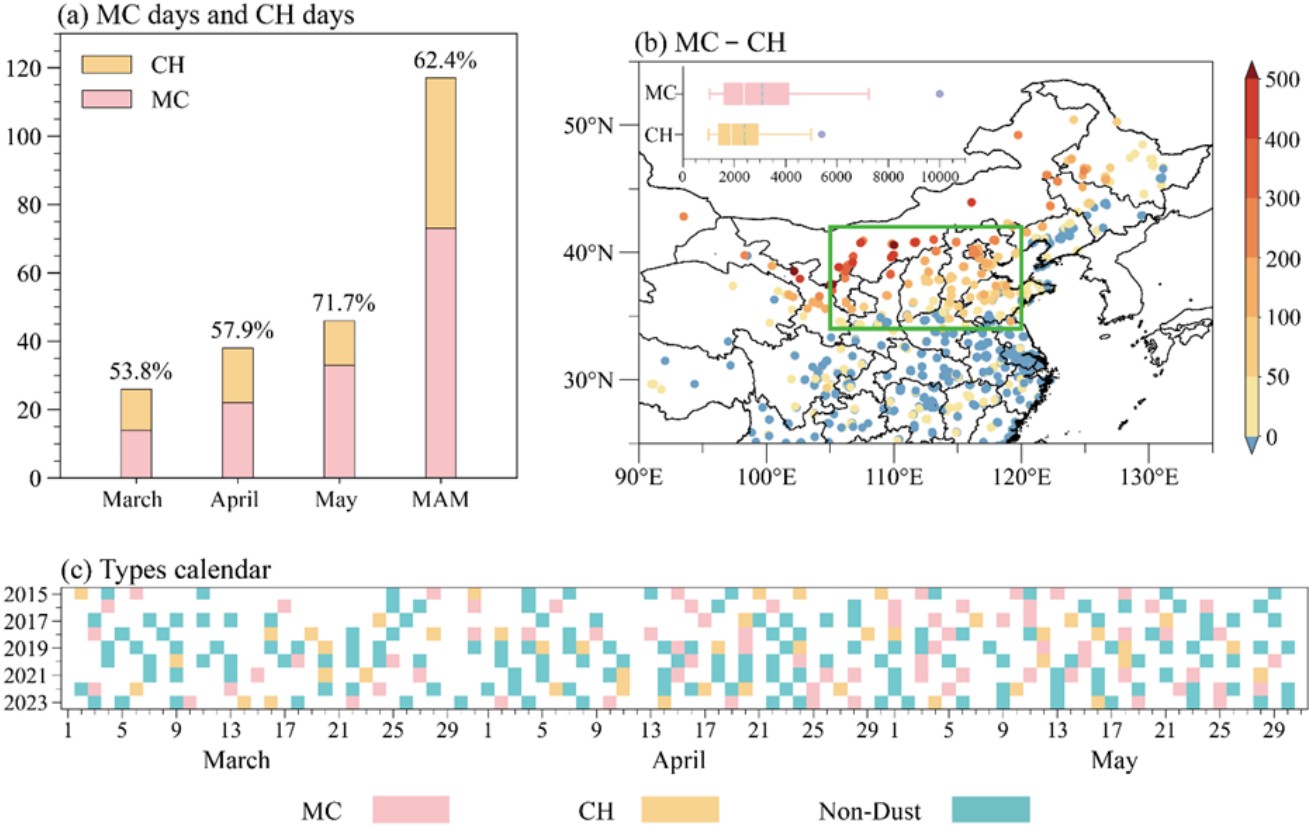

**Figure 1.** (a) The number of Dust days (stacked bar) in March, April, May and MAM (March, April and May) from 2015 to 2023. The pink parts and orange parts represent the number of MC days and CH days respectively. The proportions depicted above the bars indicate the percentages of MC days within Dust days. (b) The composite differences of observed daily maximum $PM_{10}$ concentrations (scatter, units: $\mu g\ m^{-3}$) during MC days relative to CH days. The green box indicates NC. The boxplot in the top left corner indicates daily maximum $PM_{10}$ concentrations (units: $\mu g\ m^{-3}$) of MC days (pink) and CH days (orange). The cyan dashed lines and blue dots in the boxplot represent average $PM_{10}$ concentrations and outlier values. (c) The temporal distribution of MC days, CH days and Non-Dust days in spring from 2015 to 2023.



## 4 Large-scale atmospheric circulation and associated dynamic mechanisms

In the mid-troposphere, both the MC type and CH type exhibited cyclonic anomalies located to the northwest of NC, with noticeable anticyclonic anomalies to the southeast (Fig. 2a, b). The convergence of anomalous warm and cold air masses

brought about by these cyclonic and anticyclonic anomalies in mid-latitudes led to an increase in atmospheric baroclinicity, favoring the strengthening of westerlies (Fig. 2a, b). Compared to the CH type, the cyclonic anomaly associated with the MC type was stronger and more pronounced, extending further northwestward (Fig. 2a, b). Differences in the intensity of the 500 hPa cyclonic anomalies resulted in distinct configurations of surface synoptic systems. When the 500 hPa cyclonic anomalies were stronger, as seen during MC days, the surface Mongolian cyclone dominated (Fig. S1c). This enhanced the meridional

circulation and set up a west-east orientation of the high and low-pressure systems as well as anomalies. (Fig. 3a). In contrast, for the CH days, weaker 500 hPa cyclonic anomalies led to a dominant cold high on the surface level (Fig. S1d). The meridional circulation weakened and a northwest-southeast orientation of the high and low-pressure systems as well as anomalies was set up (Fig. 3b).

The differences in system configurations could impact meteorological conditions near the surface. The pressure gradient

between high and low-pressure systems led to strong gust winds near the surface on Dust days, with the Gust10 being stronger for the MC type compared to the CH type (Fig. S4a, b). The increase in the VATD, favoring atmospheric thermodynamic instability over NC and the dust source area (Fig. S4a, b). The enhanced surface wind speeds and thermodynamic instability in the dust source area increased the turbulence intensity in the atmospheric boundary layer (Garratt, 1992). The anomalously intensified turbulence aided in entraining surface dust particles into the atmosphere (Wiggs, 2011), while also raising the PBLH

over the dust source area (Fig. S4c, d). The higher PBLH provided favorable conditions for the outward dispersion of dust particles in the air (Shao, 2008). It was observed that over the dust source area on MC and CH days, there was an anomalous divergence (Fig. 3d, e). Under the influence of anomalous northerly winds, dust particles were transported from the dust source area to NC, leading to an increase in $PM_{10}$ concentrations (Fig. 3d, e). Divergent winds not only transported dust particles outward but also inhibited the inward transport of moisture into the dust source area. This resulted in relatively dry conditions

near the surface in the dust source area, favoring dust emissions (Fig. 3d, e).

In addition to horizontal circulation, vertical circulation also played a crucial role in the generation of dust weather in NC. During Dust days, the cyclonic anomaly at 500 hPa exhibited an asymmetric vertical structure (Fig. 2d, e). On MC days, it was evident that the western ω at 500 hPa showed positive anomalies, while the eastern ω exhibited negative anomalies (Fig. 2d). The anomalous subsidence extended from the upper troposphere to the surface, transporting the westerly momentum from

the upper troposphere down to the near-surface layer (Fig. 4a), resulting in stronger V850 and Gust10 (Fig. S4a, c). With the strengthening of the westerly momentum exchange, the conversion of kinetic energy from the mean flow to turbulence was enhanced (Liu and Liu, 2011). Furthermore, the anomalous subsidence also transported dry and cold air from the middle troposphere down to the near-surface layer (Fig. 4d). The intrusion of cold air favored the disruption of the stable boundary layer, leading to atmospheric thermal instability and an increase in turbulent kinetic energy (Liu and Liu, 2011). Stronger



turbulence not only lifted surface dust but also resulted in higher PBLH. Under the influence of anomalous divergence in the lower troposphere, dust particles dispersed outward (Fig. 4d). The anomalous convergence zone in the eastern region facilitated the aggregation of dust particles and transported them to higher altitudes through anomalous upward motion. This prolonged the settling time of dust particles and facilitated their long-distance transport.

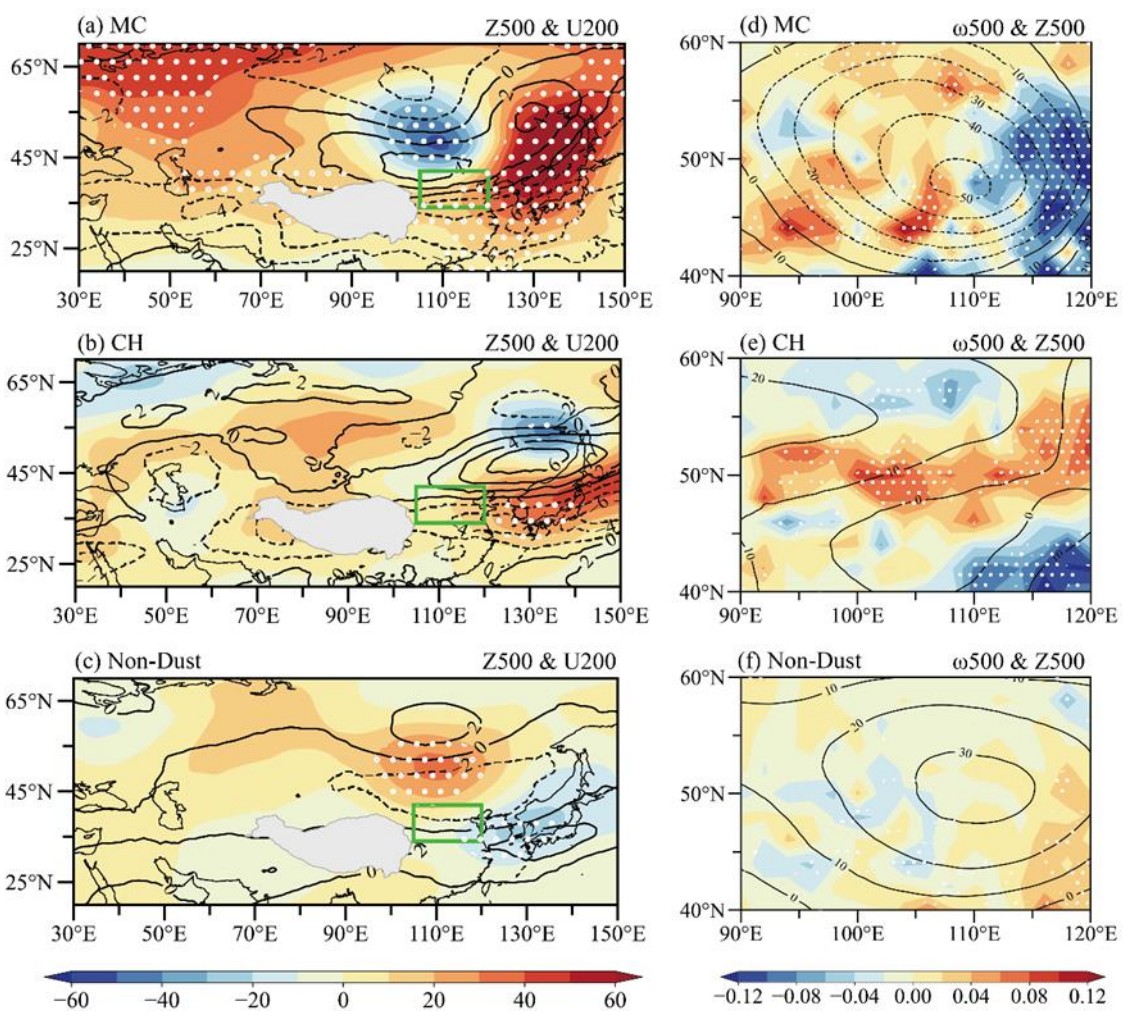

**Figure 2.** (a) Composite anomalies of Z500 (shading, units: geopotential meter, gpm) and U200 (contour, units: m s$^{-1}$) during MC days. White dots indicate that Z500 anomalies exceed the 95% confidence level. Panel (b) and (c) are the same as panel (a) but for CH days and Non-Dust days. (d) Composite anomalies of ω500 (shading, units: Pa s$^{-1}$) and Z500 (contour, units: gpm) during MC days. White dots indicate that ω500 anomalies exceed the 95% confidence level. Panel (e) and (f) are the same as panel (d) but for CH days and Non-Dust days. The green boxes in panel (a)–(f) represent NC.





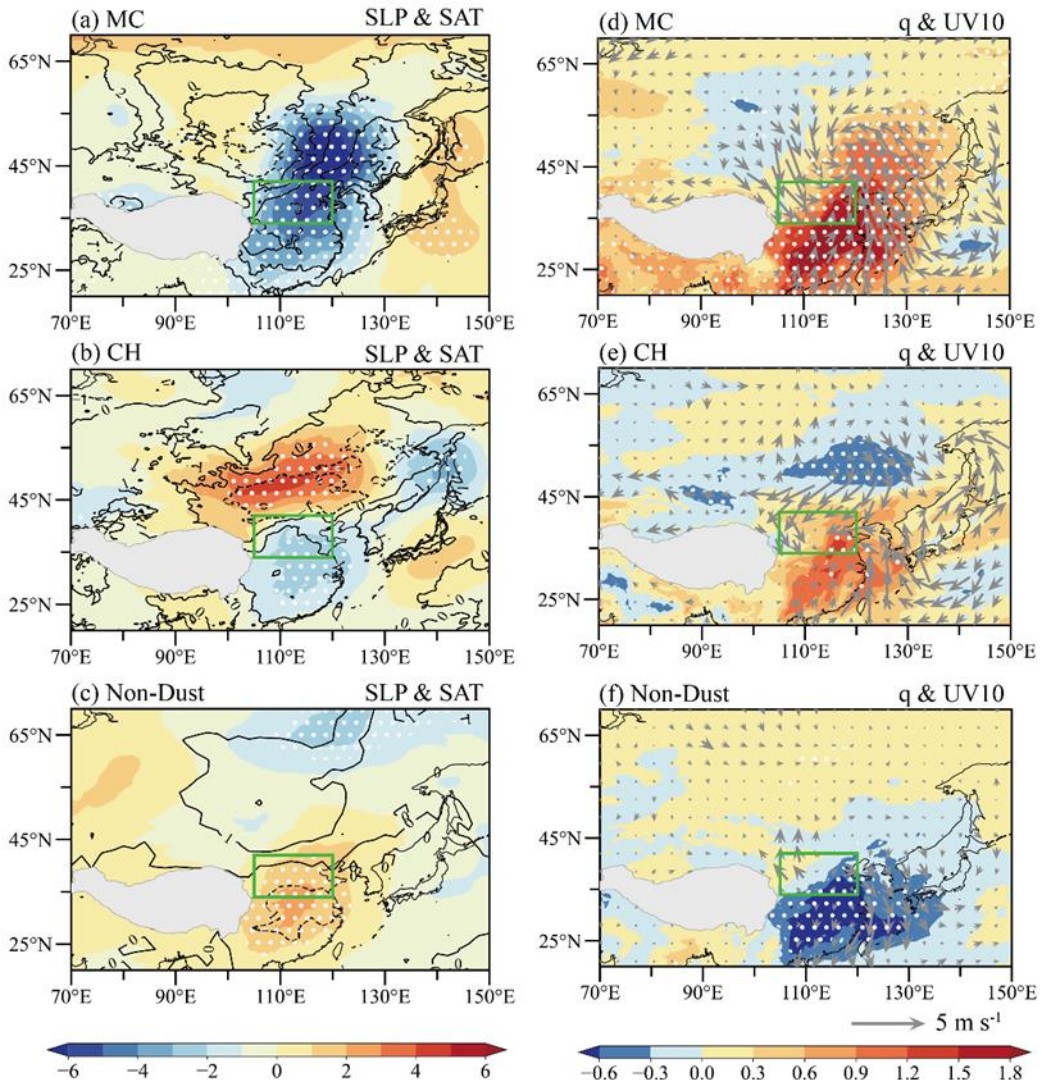

**Figure 3.** (a) Composite anomalies of SLP (shading, units: hPa) and SAT (contour, units: K) during MC days. White dots indicate that SLP anomalies exceed the 95% confidence level. Panel (b) and (c) are the same as panel (a) but for CH days and Non-Dust days. (d) Composite anomalies of q (shading, units: $10^{-3}$ kg kg$^{-1}$) and UV10 (vectors, units: m s$^{-1}$) during MC days. White dots indicate that q anomalies exceed the 95% confidence level. Panel (e) and (f) are the same as panel (d) but for CH days and Non-Dust days. The green boxes in panel (a)–(f) represent NC.

The latitudinal contrast of the CH type anomalous vertical circulation was weaker than that of the MC type (Fig. 4b, e), and the 500 hPa vertical velocity anomaly exhibited a northwest-southeast distribution pattern (Fig. 2e). The enhanced anomalous meridional vertical circulation increased the importance of the dust source area in the northward direction outside NC (Fig S5). The vegetation coverage in the northerly direction was relatively better than that in the northwesterly direction in the dust source area (Fig. S3a), leading to lower PM$_{10}$ concentrations in NC under the influence of the CH type compared





to the MC type. Overall, the meteorological variables on CH days exhibited spatial distribution characteristic similar to MC days but with weaker intensity. These results partially explained why $PM_{10}$ concentrations were relatively lower on CH days compared to MC days.

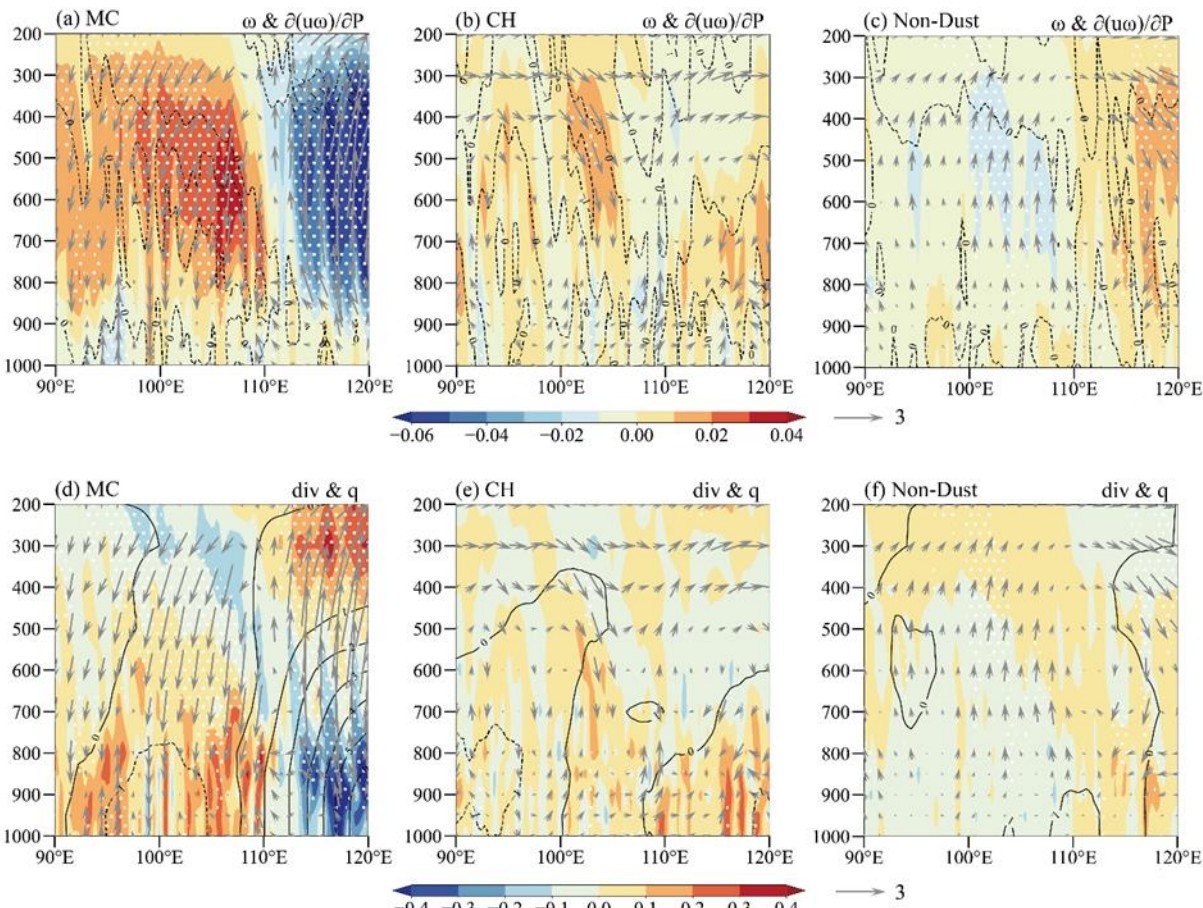

**Figure 4.** (a) Composite anomalies of zonal component of the vertical circulation average over 40–60°N, 90–120°E during MC days. The variables include ω (shading, units: Pa s$^{-1}$) and downward transport of westerly momentum (<0, dashed contour, units: $10^{-3}$ m s$^{-2}$). White dots indicate that ω anomalies exceed the 95% confidence level. The vectors represent ω (magnified 100 times) and zonal wind. Panel (b) and (c) are the same as panel (a) but for CH days and Non-Dust days. (d) Composite anomalies of zonal component of the vertical circulation average over 40–60°N, 90–120°E during MC days. The variables include divergence (shading, units: $10^{-5}$ s$^{-1}$) and q (contour, units: $10^{-4}$ kg kg$^{-1}$). White dots indicate that divergence anomalies exceed the 95% confidence level. The vectors represent ω (magnified 100 times) and zonal wind. Panel (e) and (f) are the same as panel (d) but for CH days and Non-Dust days.

For comparison, the atmospheric circulation and meteorological conditions on Non-Dust days were also analyzed. The atmospheric circulation anomalies on Non-Dust days exhibited distribution characteristics opposite to those on Dust days. In the middle troposphere, the anticyclonic and cyclonic anomalies impeded the southward movement of cold air and the




northward movement of warm air (Fig. 2c). The atmospheric baroclinicity weakened leading to a negative anomaly in the
westerly wind (Fig. 2c). The opposite surface circulation anomalies led to southerly wind anomalies (Fig. 3c, f), causing an
increase in q in the dust source area (Fig. 3f). During Non-Dust days, there was an opposite vertical circulation characterized
by anomalous upward motion in the western region and anomalous downward motion in the eastern region (Fig. 2f). The
anomalous upward motion suppressed the downward momentum flux and the intrusion of dry and cold air from the upper
levels (Fig. 4c, f). All the meteorological conditions on Non-Dust days were unfavorable for the occurrence of dust weather.

## 5 A common predictor for the two dust weather types

Based on the results of Sect. 4, correlations between meteorological conditions and the daily maximum $PM_{10}$
concentration in NC can be observed. To further confirm the relationship, the spatial correlation coefficients were calculated
(Fig. S6). Significant correlation coefficients between meteorological conditions and $PM_{10}$ concentration further confirmed
the aforementioned results (Fig. S6). The increased meridional temperature differences reflected the enhanced atmospheric
baroclinicity in mid-latitude regions (Fig. S6f; Zhang et al., 2021), which provided favorable conditions for the generation of
surface synoptic systems and strong wind anomalies. In the selected regions with significant correlations, corresponding
meteorological indices were defined, and the correlation coefficients between the daily maximum $PM_{10}$ concentration in NC
were calculated (Table 1). All correlation coefficients passed a significance test at the 99% level (p<0.01). By examining the
area averaged meteorological variables in the regions of significant correlation, it was also found that the meteorological
conditions on MC days, CH days as well as Non-Dust days, exhibited significant differences and had certain correlations with
the $PM_{10}$ concentration (Fig. 5b–i). Specifically, the meteorological conditions favoring dust weather in the MC type were
stronger than those in the CH type, leading to higher $PM_{10}$ concentrations on MC days compared to CH days, making extreme
dust events more likely to occur on MC days.

As described in Sect. 4, the 500 hPa cyclonic anomaly affected near-surface meteorological conditions through the
associated vertical and horizontal circulation anomalies, leading to dust weather in NC. By correlation analysis, it was found
that there was a significant relationship between 500 hPa cyclonic anomaly (I_Z500c) and the related meteorological condition
indices (Table 2). This confirmed the physical mechanisms outlined in Sect. 4. However, based on the correlation coefficient
results, 500 hPa cyclonic anomaly (I_Z500c) alone did not explain the anomalies in VATD and PBLH well (Table 2). At the
500 hPa level, there was also an anticyclonic anomaly that showed a significant correlation with the $PM_{10}$ concentration in NC
(Fig. S6a). This anomaly provides warm air advection disturbances in the mid-latitudes on Dust days. Therefore, an analysis
of its relationship with near-surface meteorological conditions was warranted. The correlation coefficients between the
anticyclonic anomaly (I_Z500a) and the near-surface meteorological indices all passed a significance test at the 99% level
(Table 2). The strongest positive correlations were observed between I_Z500a and both I_VATD and I_PBLH, with correlation
coefficients reaching 0.639 and 0.534, respectively (Table 2).





**Table 1.** The definition of meteorological indices related to dust weather in NC and correlation coefficients (R) of observed daily maximum PM$_{10}$ concentrations over NC with each index in spring from 2015 to 2023. All indices were the normalized area-averaged corresponding variables for their corresponding areas. All the correlation coefficients exceed the 99% confidence level.

| Index | Definition | R |
|:---:|:---:|:---:|
| I_Z500c | Z500 over (44–52°N, 98–113°E) | -0.205 |
| I_Z500a | Z500 over (33–40°N, 123–137°E) | 0.167 |
| I_ω500 | ω500 over (42–48°N, 97–107°E) | 0.157 |
| I_U200 | U200 over (40–45°N, 100–115°E) | 0.220 |
| I_V850 | V850 over (38–46°N, 102–110°E) | -0.270 |
| I_Gust10 | Gust10 over (37–49°N, 103–118°E) | 0.355 |
| I_SAT | SAT over (23–33°N, 105–120°E) minus SAT over (43–50°N, 100–110°E) | 0.383 |
| I_q | q over (45–53°N, 91–103°E) | -0.171 |
| I_VATD | VATD over (42–47°N, 106–116°E) | 0.118 |
| I_PBLH | PBLH over (46–55°N, 100–110°E) | 0.126 |
| I_ACA-CA | Z500 over (33–40°N, 123–137°E) minus Z500 over (44–52°N, 98–113°E) | 0.321 |

**Table 2.** The correlation coefficients of I_Z500c, I_Z500a and I_ACA-CA with other meteorological indices (Table 1) in spring from 2015 to 2023. All the correlation coefficients exceed the 99% confidence level. The symbol "—" denotes that the correlation coefficient is not statistically significant and therefore has been excluded from the analysis.

| Index | I_Gust10 | I_VATD | I_PBLH | I_SAT | I_q | I_ω500 | I_U200 | I_V850 |
|:---:|:---:|:---:|:---:|:---:|:---:|:---:|:---:|:---:|
| I_Z500c | -0.216 | 0.500 | 0.191 | -0.462 | 0.693 | -0.130 | -0.470 | 0.195 |
| I_Z500a | 0.256 | 0.639 | 0.534 | 0.292 | 0.367 | — | 0.269 | -0.117 |
| I_ACA-CA | 0.407 | 0.113 | 0.292 | 0.652 | -0.287 | — | 0.638 | -0.269 |

The common circulation index I_ACA-CA of the two dust weather types was defined by calculating the difference in Z500 between the 500 hPa anticyclonic anomaly and cyclonic anomaly and normalizing it (Table 1). Significant differences in composite I_ACA-CA were observed for MC days, CH days, and Non-Dust days (Fig. 5a). These differences corresponded to the relationship with PM$_{10}$ concentrations among the three types. I_ACA-CA exhibited significant correlations with PM$_{10}$ concentrations, showing a correlation coefficient of 0.321, passing the 99% significance test (p<0.01). It also demonstrated correlations with near-surface meteorological indices and horizontal circulation indices (Table 2). These correlation coefficients effectively confirmed that the 500 hPa cyclonic and anticyclonic circulation anomalies influenced near-surface meteorological conditions through horizontal and vertical circulation. Figure 6 displays the daily I_ACA-CA values in spring from 2015 to 2023, where positive I_ACA-CA captured 83.6% of MC days, 63.6% of CH days, and 76.1% of Dust days





overall. All the Dust days in 2016 and 2023 were captured by I_ACA-CA (Fig. 6). In 2021, only one instance of lower $PM_{10}$ concentration was not captured, with the rest being captured (Fig. 6). In correspondence with the positive I_ACA-CA observed two days, one day (I_ACA-CA>its one standard deviation), and zero day (I_ACA-CA>its one standard deviation) in advance,

successful capture rates of 55.6%, 69.2%, and 76.1% for Dust days were achieved. These high percentages suggest that the reinforced positive I_ACA-CA significantly contributed to the high $PM_{10}$ concentrations in NC. Thus, I_ACA-CA served as a meaningful indicator for forecasting dust weather in NC.

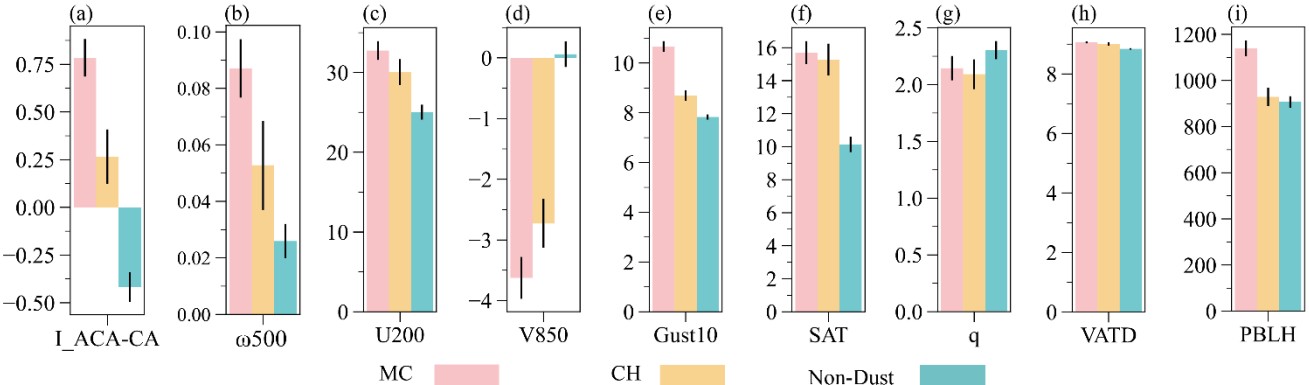

**Figure 5.** Composite meteorological index: (a) I_ACA-CA, and composite meteorological variables (composite values of corresponding

meteorological indices before normalization in Table 1): (b) ω500 (unit: Pa s$^{-1}$), (c) U200 (unit: m s$^{-1}$), (d) V850 (unit: m s$^{-1}$), (e) Gust10 (unit: m s$^{-1}$), (f) SAT (unit: K), (g) q (unit: kg kg$^{-1}$), (h) VATD (unit: K), (i) PBLH (unit: m) during MC days, CH days and Non-Dust days. The black error bars indicate the standard error.

## 6 Conclusion and discussion

In recent years, the frequency and intensity of dust weather in NC have been increasing, with many adverse effects on

human health, national economy, and ecological environment. This study selected Dust days based on $PM_{10}$ concentrations in NC during spring from 2015 to 2023. By objectively identifying the presence of the Mongolian cyclone and based on the main surface synoptic systems, Dust days were classified into MC type (62.4%) and CH type (37.6%). The $PM_{10}$ concentrations on MC days were higher and more extreme compared to those on CH days, with average $PM_{10}$ concentrations at 3076 μg m$^{-3}$ and 2391 μg m$^{-3}$. The research results indicate that the 500 hPa atmospheric circulation anomalies affect near-surface

meteorological conditions through related horizontal and vertical circulation, leading to different intensities of dust weather in NC. Figure 7 illustrates the three-dimensional atmospheric circulation anomalies structures and relevant dynamic processes of the two types of dust weather with distinct $PM_{10}$ concentrations in NC. The different intensities of the 500 hPa cyclonic and anticyclonic anomalies led to differences in the configuration of surface systems. This was manifested as different meridional extents of circulation, with the meridional circulation of the MC type being greater than that of the CH type. This also resulted

in differences in the intensity of near-surface meteorological factors, with the meteorological conditions for dust weather being




more favorable for the MC type compared to the CH type. The strong gust winds and atmospheric thermal instability caused dust particles from the dust source areas to be lifted into the air, with the anomalous downward momentum in the upper troposphere exacerbating this process. Subsequently, under the influence of the anomalous northerly winds, dust particles were transported to NC, leading to an increase in $PM_{10}$ concentrations, with the $PM_{10}$ concentration on MC days higher than that on

CH days.

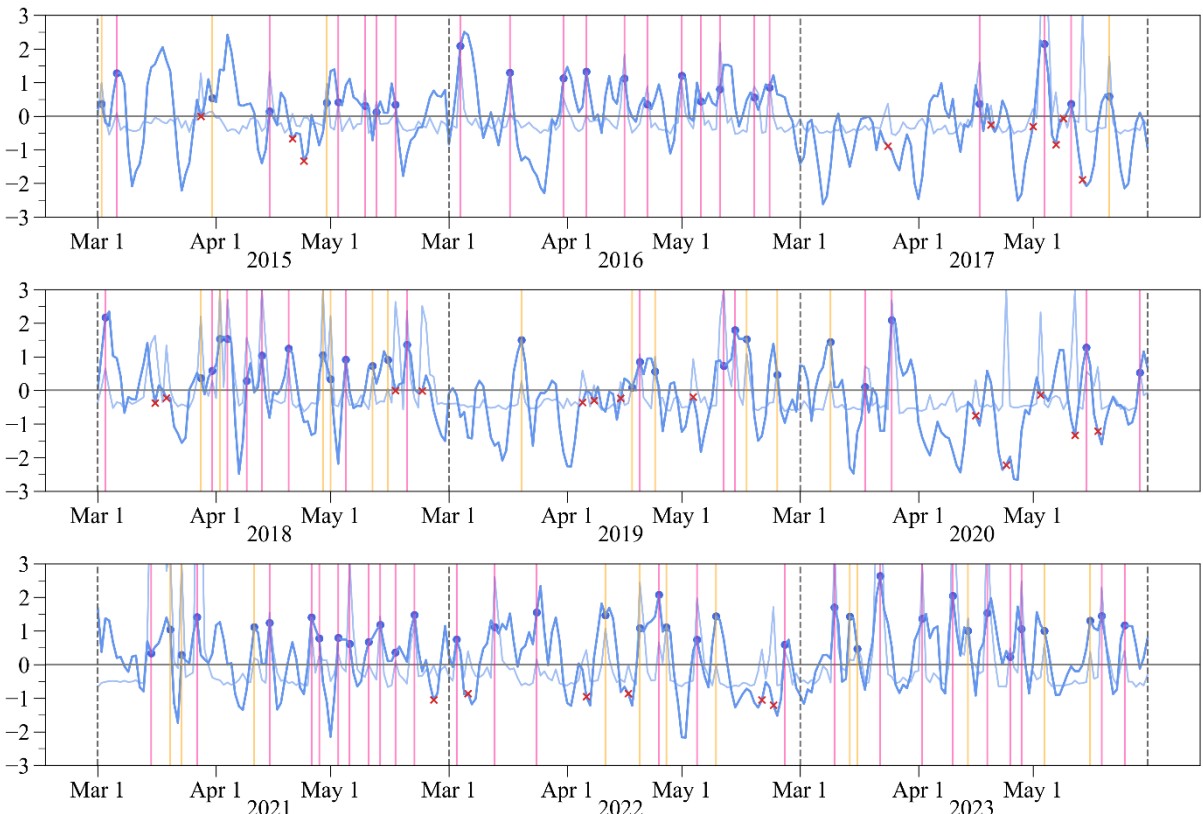

**Figure 6.** Daily I_ACA-CA (blue line) and normalized daily maximum $PM_{10}$ concentrations observed in NC (light blue line) in March (Mar), April (Apr) and May from 2015 to 2023. The blue dots indicate the Dust days captured when I_ACA-CA>0 and the corresponding I_ACA-CA value. The pink and orange vertical lines indicate the MC days and CH days captured when I_ACA-CA>0, respectively. The red "x"

marks represent the Dust days that I_ACA-CA failed to capture. The gray vertical dashed lines separate each year.

The common predictor (I_ACA-CA) of the two dust weather types in NC was identified. The research findings demonstrated that the I_ACA-CA had good indicative significance for dust weather in NC. The ability of the C3S model (ECMWF SEAS5.1) to reproduce I_ACA-CA was further assessed. The I_ACA-CA calculated by C3S data with a one-month lead captured 47.0% of spring Dust days when positive. The relative error compared to the capture rate (76.1%) of I_ACA-

CA calculated by ERA5 data was 38.2%. The I_ACA-CA calculated from C3S model forecast data has a lower capture rate for dust weather and exhibits errors compared to ERA5, indicating the need for further improvement. It is worth noting that



due to the lower spatial resolution (1°×1°) of the C3S model forecast data relative to the ERA5 data (0.25°×0.25°), the SLP produced by the C3S model failed to effectively identify the presence of the Mongolian cyclone. Therefore, the introduction of the common predictor (I_ACA-CA) is of great significance for dust weather prediction in NC. However, due to constraints

imposed by $PM_{10}$ concentration observation data, the study period only covered the years 2015 to 2023. Further research is needed to test the effectiveness of the I_ACA-CA indicator over longer time series and to utilize it for predicting future dust weather. The study solely considered the impact of dynamic factors on dust weather. While the dust source condition also plays a crucial role in the formation of dust weather. Previous research by Yin et al. (2022) found that climate accumulation effects could influence the dust source conditions for spring dust weather, thereby regulating its intensity. For the "3.15" super

dust storm in 2021, although the I_ACA-CA exhibited a certain level of positive signal, it did not indicate the extremity of the dust weather event (Fig. 6). For the severe dust storm on March 22 2023, the I_ACA-CA indicator performed well, showing the extreme characteristics in intensity (Fig. 6). The deviation in indicating extreme dust storm intensity by I_ACA-CA may be due to the lack of consideration of dust source conditions. Therefore, the development of a dust weather prediction model that comprehensively considers the synergistic effects of dust source conditions and dynamic factors awaits further research.

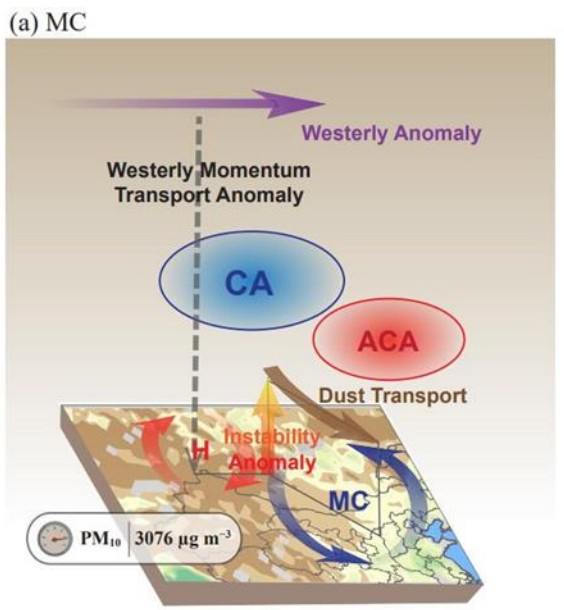
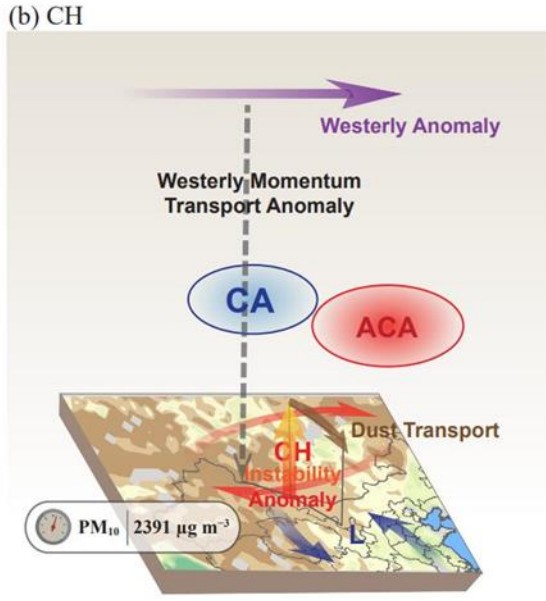


**Figure 7.** Schematic diagram for the three-dimensional atmospheric circulation anomalies and related dynamic processes of (a) MC type and (b) CH type dust weather with distinct $PM_{10}$ concentrations in NC. 500 hPa cyclonic anomaly (CA) and anticyclonic anomaly (ACA) led to different configurations of anomalous circulation systems on the surface level. The anomalous gust winds and thermal instability near the dust source area favored dust lifting. Enhanced 200 hPa westerly winds, with momentum transport downward, favored further increases

in surface wind speeds. Anomalous northerly winds transported dust particles southward affecting NC. The shading on the surface represents NDVI in March 2023. The directions of the arrows indicate anomalous airflow directions. The average $PM_{10}$ concentrations of MC and CH days are demonstrated in the left bottom of each panel.



**Data Availability.**

Hourly $PM_{10}$ concentration data was available at https://quotsoft.net/air/ (China National Environmental Monitoring Centre,

2023). Hourly ERA5 reanalysis dataset was available at https://cds.climate.copernicus.eu/cdsapp#!/dataset/reanalysis-era5-pressure-levels?tab=overview (ERA5, 2024). The NDVI data could be acquired from https://www.ncei.noaa.gov/access/metadata/landing-page/bin/iso?id=gov.noaa.ncdc:C01558 (NOAA, 2023). The C3S seasonal forecast data was available at https://cds.climate.copernicus.eu/cdsapp#!/dataset/10.24381/cds.50ed0a73?tab=overview (C3S, 2024).

**Authors' contribution**

Yin Z. C. and Wang H. J. designed the research. Huo Q. Y., Ma X. Q. and Yin Z. C. performed the research. Huo Q. Y. prepared the manuscript with contributions from all co-authors.

**Competing interests**

The authors declare that they have no conflict of interest.

**Acknowledgements**

This work was supported by the National Natural Science Foundation of China (42088101).





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
