# Peer review of "Distinctive dust weather intensities in North China resulted from two types of atmospheric circulation anomalies"

_EGUsphere, 2024_

## Author Response (AR2)

**Reply to Editor:**

Dear authors,

before publishing the article, please complete the data acknowledgement section by including all data links to websites in English.

Sincerely,
Stephanie Fiedler

*Reply:*

Thank you for your efforts and guidance. The **data links** to websites in English in the data availability section have been **corrected**. As the website for $PM_{10}$ and $PM_{2.5}$ concentration data (https://quotsoft.net/air/) is in Chinese, relevant **translations** are provided for reference (Fig. R1). The national air quality data on the website is sourced from the National Urban Air Quality Real-time Publishing Platform of the China Environmental Monitoring Station, updated daily. The English translation of this platform is also provided (Fig. R2).

*Revision:*

**p. 17, line 350-355:**

**Data Availability**

All data used in this article can be publicly downloaded. Hourly $PM_{10}$ and $PM_{2.5}$ concentration data was available at https://quotsoft.net/air/ (China National Environmental Monitoring Centre, 2024). Hourly ERA5 reanalysis dataset was available at https://cds.climate.copernicus.eu/datasets/reanalysis-era5-pressure-levels?tab=download (ERA5, 2024a) and https://cds.climate.copernicus.eu/datasets/reanalysis-era5-single-levels?tab=download (ERA5, 2024b). The NDVI data could be acquired from https://www.ncei.noaa.gov/access/metadata/landing-page/bin/iso?id=gov.noaa.ncdc:C01558 (NOAA, 2023). The C3S seasonal forecast data was available at https://cds.climate.copernicus.eu/datasets/seasonal-original-pressure-levels?tab=download (C3S, 2024a) and https://cds.climate.copernicus.eu/datasets/seasonal-original-single-levels?tab=download (C3S, 2024b).

[Figure]

本站提供

中国空气质量历史数据下载（2014/05/13以来）

北京市空气质量历史数据下载（2013/12/06以来）

中国气象历史数据下载（1942/07以来）

欢迎各种分析、研究

空气质量数据类型包括PM2.5, PM10, SO2, NO2, O3, CO, AQI。

全国空气质量数据来自中国环境监测总站的全国城市空气质量实时发布平台，每日更新。

北京市空气质量数据来自北京市环境保护检测中心网站，每日更新。

气象数据要素包括气温、气压、露点、风向风速、云量、降水量。

气象数据来自美国国家气候数据中心（NCDC），每年不定期更新。

This site provides:

•Download of historical air quality data for China (since May 13, 2014)

•Download of historical air quality data for Beijing (since December 6, 2013)

•Download of historical meteorological data for China (since July 1942)

•Welcome for various analyses and research

Air quality data types include PM2.5, PM10, SO2, NO2, O3, CO, AQI.

National air quality data is sourced from the National Urban Air Quality Real-time Publishing Platform of the China Environmental Monitoring Station, updated daily.

Beijing air quality data is sourced from the website of the Beijing Environmental Protection Monitoring Center, updated daily.

Meteorological data elements include temperature, pressure, dew point, wind speed and direction, cloud cover, and precipitation.

Meteorological data is sourced from the National Climatic Data Center (NCDC) of the United States, updated irregularly throughout the year.

**单日空气质量数据下载地址**

| 全国国控监测点数据 CSV格式 | https://quotsoft.net/air/data/china_sites_[日期].csv |
| 全国城市数据 CSV格式 | https://quotsoft.net/air/data/china_cities_[日期].csv |
| 北京PM2.5/PM10/AQI数据 CSV格式 | https://quotsoft.net/air/data/beijing_all_[日期].csv |
| 北京SO2/NO2/O3/CO数据 CSV格式 | https://quotsoft.net/air/data/beijing_extra_[日期].csv |

其中"[日期]"为8位数字表示的日期，例如2013/12/05的数据地址为 https://quotsoft.net/air/data/beijing_all_20131205.csv。

**Single-day air quality data download links:**

National key monitoring station data in CSV format: https://quotsoft.net/air/data/china_sites_[date].csv

National city data in CSV format: https://quotsoft.net/air/data/china_cities_[date].csv

Beijing PM2.5/PM10/AQI data in CSV format: https://quotsoft.net/air/data/beijing_all_[date].csv

Beijing SO2/NO2/O3/CO data in CSV format: https://quotsoft.net/air/data/beijing_extra_[date].csv

Where "[date]" is represented by an 8-digit numerical date, for example, the data link for December 5, 2013, would be https://quotsoft.net/air/data/beijing_all_20131205.csv

**Figure R1.** Translation of the important content of the website for $PM_{10}$ and $PM_{2.5}$ concentration data (https://quotsoft.net/air/).

[Figure]

**Figure R2.** Translation of the National Urban Air Quality Real-time Publishing Platform (https://air.cnemc.cn:18007).

**Reply to Referee 1:**

The study evaluated the intensity of dust weather from $PM_{10}$ concentrations and identified the synoptic systems and related dynamic mechanisms that caused different intensities of dust weather in North China. In addition to the well-known Mongolian cyclone that had received much attention in recent years, the Mongolian cold high was also responsible for dust weather in North China. Considering both the Mongolian cyclone and the cold high for forecasting, a common predictor was proposed. The results of this study could provide references for the forecasting of dust weather and climate prediction. **This paper is well written and organized. I recommend it to be published in ACP after several minor corrections.**

Major comments:

1. In Section 2.2, the identification method of the Mongolian cyclone was described based on its definition, but the description was not very specific. Could further details be provided?

*Reply:*

The method used in this article to identify the Mongolian cyclone is based on the meteorological definition of the extratropical cyclone (Shou, 2006). The **specific identification steps** are as follows: First, locate the lowest sea level pressure (SLP) within the range of **40–55°N, 100–130°E**. If the **lowest SLP is less than or equal to 1010 hPa**, then proceed to calculate the average value of the **pressure gradient within a range of ±2.5° latitude and longitude around the lowest SLP**. If the average pressure gradient is greater than or equal to 0.55 hPa per 100 km, the presence of the Mongolian cyclone is confirmed; otherwise, the Mongolian cyclone is considered not to exist. The more precise description of the method for identifying Mongolian cyclones has been revised.

*Related References:*

Shou S. W.: Synoptic Analysis, China Meteorological Press, Beijing, 361 pp., ISBN 9787502934576, 2006 (in Chinese).

*Revision:*

**p. 3, line 90–93:** According to the synoptic definition of the extratropical cyclone (Shou, 2006), the Mongolian cyclone was identified based on the following criteria: (1)

The **lowest SLP** within the range of **40–55°N, 100–130°E** should not exceed **1010 hPa.**

(2) The **average pressure gradient** within a **±2.5° latitude and longitude range** around the lowest SLP must be equal to or greater than **0.55 hPa per 100 km**. The vertical air temperature …

2. Section 5 focused on the common predictor of the MC type and the CH type. However, the improvement and advantage of this common predictor, compared to solely considering the Mongolian cyclone, are not clearly articulated in the text. It is recommended to provide further elaboration on this point to enhance clarity and understanding.

*Reply:*

Previous studies have generally highlighted the significant role of Mongolian cyclones in dust weather in North China (Wu et al., 2016; Bueh et al., 2022; Gao et al., 2024). This study emphasized the role of other systems, mainly cold high, in addition to Mongolian cyclones. By solely focusing on Mongolian cyclones, the influence of **other systems** on North China's dust weather (accounted for **38.3%**) would be **overlooked**. Based on ERA5 reanalysis data from 2015 to 2023, the dust capture rate of the common predictor is **76.5%**, which **captures more dust days** compared to solely considering the Mongolian cyclone (**61.7%**). Furthermore, the ability of the **C3S seasonal forecast model to reproduce I_ACA-CA** was further assessed. The I_ACA-CA calculated by ECMWF, DWD, and MF seasonal forecast models with a one-month lead captured around 50% of spring dust days when positive. It is worth noting that due to the lower spatial resolution (1°×1°) of the C3S model forecast data relative to the ERA5 data (0.25°×0.25°), the **SLP produced by the C3S model failed to effectively identify the presence of the Mongolian cyclone**. Therefore, the introduction of the common predictor (I_ACA-CA) is of great significance for dust weather prediction in NC. In the discussion section, explanations of the advantages of the common predictor over solely considering Mongolian cyclones were added.

*Related References:*

Bueh, C., Zhuge, A., Xie, Z., Yong, M., and Purevjav, G.: The development of a

powerful Mongolian cyclone on 14–15 March 2021: Eddy energy analysis, AOSL., 15, 100259, https://doi.org/10.1016/j.aosl.2022.100259, 2022.

Gao, J., Ding, T., and Gao, H.: Dominant circulation pattern and moving path of the Mongolian Cyclone for the severe sand and dust storm in China, Atmos. Res., 301, 107272, https://doi.org/10.1016/j.atmosres.2024.107272, 2024.

Wu, C. L., Lin, Z. H., He, J. X., Zhang, M. H., Liu, X. H., Zhang, R. J., and Brown, H.: A process-oriented evaluation of dust emission parameterizations in CESM: Simulation of a typical severe dust storm in East Asia, J. Adv. Model. Earth Syst., 8, 1432–1452, C10.1002/2016MS000723, 2016.

*Revision:*

**p. 13, line 292:** …The common predictor offers a more **comprehensive prediction** for both types of dust weather compared to solely considering the Mongolian cyclone, capturing more dust days. The ability of the C3S seasonal forecast model…

3. In Section 6, the ability of the C3S model to reproduce I_ACA-CA was discussed, but only the ECMWF SEAS5.1 was considered. Why was only the predictive ability of one model considered? Is there a certain degree of randomness involved? It is recommended to also compare and evaluate the capabilities of other systems.

*Reply:*

Thank you for your suggestions. Among C3S models, only **ECMWF SEAS5.1 has continuous data for Z500 from 2015 to 2023**. **Deutscher Wetterdienst (DWD) and Météo-France** have data for 2015-2023 **but from different system versions**, while **other institution systems have missing data for certain years**. Therefore, we conducted additional analysis of the capabilities of the **DWD forecast systems (GCFS2.0 & GCFS2.1)** and **Météo-France forecast systems (System6 & System7 & System8)** to reproduce I_ACA-CA, and compared them with ECMWF SEAS5.1.

Although there may be some deviations when using data from different versions of systems simultaneously, we still utilized forecast data from the DWD and MF systems to calculate I_ACA-CA for comparison. The I_ACA-CA calculated by **ECMWF**, **DWD, and MF seasonal forecast models** with a one-month lead captured **46.1%**, **52.2%, and 51.3%** of spring dust days when positive. The capture rates are all **around 50%**, indicating that using only one model has **no randomness**. The **discussion** on the **ability of DWD and MF** seasonal forecast models to reproduce

I_ACA-CA has been added.

*Revision:*

**p. 3, line 84-86:** … Environmental Information (Vermote, 2019). The Copernicus Climate Change Service (C3S, 2018) provided seasonal forecast products from European Centre for Medium-Range Weather Forecasts (ECMWF) SEAS5.1, **Deutscher Wetterdienst (DWD) GCFS2.0 & GCFS2.1**, and **Météo-France (MF) System6 & System7 & System8**. In this study …

**p. 13, line 292-295:** … The ability of the C3S seasonal forecast model to reproduce I_ACA-CA was further assessed. The I_ACA-CA calculated by ECMWF, **DWD, and MF seasonal forecast models** with a one-month lead captured **around 50%** of spring dust days when positive.

Specific comments:

1. Lines 114-115: The sentence: "the main surface synoptic systems for the two types of Dust days were the Mongolian cyclone and cold high" is ambiguous. "According to the context of the text, it is proposed to be modified as: "the main surface synoptic systems for the two types of Dust days were the Mongolian cyclone and cold high respectively".

*Reply:*

Thank you for your advice. This sentence has been **revised** according to the suggestion.

*Revision:*

**p. 4, line 114-115:** … the main surface synoptic systems for the two types of Dust days were the Mongolian cyclone and cold high respectively …

2. The abstract states that the Mongolian cyclone type accounts for 62.4%, with the remaining 37.6% being the cold high type. However, based on Fig. 1, it seems like both of the types together make up 62.4%. The percentages labeling in Fig. 1 are misleading. It is recommended to make corrections.

*Reply:*

Thank you for your advice. In order to avoid confusion, the percentages in Fig. 1 have been **removed**.

*Revision:*

**p. 5, line 134-140:**

[Figure]

**Figure 1.** (a) Boxplots of daily maximum $PM_{10}$ concentrations (units: $\mu g\ m^{-3}$) in NC during MC days (pink) and CH days (orange). The cyan dashed lines and blue dots in the boxplot represent average $PM_{10}$ concentrations and outlier values. Density distributions of $PM_{10}$ concentrations are shown by pink and orange shadings for MC days and CH days respectively. (b) The composite differences of observed daily maximum $PM_{10}$ concentrations (scatter, units: $\mu g\ m^{-3}$) during MC days relative to CH days. The green box indicates NC.

3. Based on the content in the main text, the meteorological indices in Table 1 are calculated corresponding to the area with the most significant correlation coefficients with the daily maximum $PM_{10}$ concentrations. It is recommended that, the corresponding regions where the indices are calculated should be clearly marked on the map to make the definition of the indices more explicit and clearer.

*Reply:*

To provide a more intuitive display of the corresponding regions for calculating the meteorological indices, these areas have been marked with black boxes.

[Figure]

**Figure R1.** Correlation coefficients of observed daily maximum $PM_{10}$ concentrations over NC with daily (a) Z500, (b) ω500, (c) U200, (d) V850, (e) Gust10, (f) SAT, (g) q, (h) PBLH, and (i) VATD in spring from 2015 to 2023. White dots indicate that correlation coefficients exceed the 95% confidence level. The green boxes in panel (a)–(i) represent NC. The black boxes in panel (a)–(i) represent the regions for calculating the indices in Table 1 respectively.

4. The "L" and "H" in Fig. 7 are not explained in the caption, please add clarification.

*Reply:*

The descriptions of the meanings of "L" and "H" have been added to the caption of the figure.

*Revision:*

**p. 14, line 310-317:**

[Figure]

[Figure]

**Figure 8.** Schematic diagram for the three-dimensional atmospheric circulation anomalies and related dynamic processes of (a) MC type and (b) CH type dust weather with distinct $PM_{10}$ concentrations in NC. 500 hPa cyclonic anomaly (CA) and anticyclonic anomaly (ACA) are the key anomalous circulation systems for the two types. **"L" and "H"** respectively represent **surface low-pressure anomalies** and **high-pressure anomalies**. The anomalous gust winds and thermal instability near the dust source area favored dust lifting. Enhanced 200 hPa westerly winds, with momentum transport downward, favored further increases in surface wind speeds. Anomalous northerly winds facilitated the emission and transport of dust particles. The shading on the surface represents NDVI in March 2023. The directions of the arrows indicate anomalous airflow directions. The average $PM_{10}$ concentrations of MC and CH days are demonstrated in the left bottom of each panel.

5. Line 318: The period after the subheading should be removed.

*Reply:*

The period after the subheading has been removed.

*Revision:*

**p. 15, line 318:** Data Availability

6. Line 36 in Supplement: There is an error in the caption of Fig. S5: "zonal wind" should be "meridional wind".

*Reply:*

This error has been revised. Composite anomalies of meridional component of the vertical circulation during CH days in the supplement have been moved to Fig. 4 in the revised manuscript.

*Revision:*

[Figure]

**Figure 4.** Composite anomalies of zonal component of the vertical circulation average over 40–60°N, 90–120°E during MC days: (a) The variables include ω (shading, units: Pa s$^{-1}$) and downward transport of westerly momentum (<0, dashed contour, units: $10^{-3}$ m s$^{-2}$). White dots indicate that ω anomalies exceed the 95% confidence level. The vectors represent ω (magnified 100 times) and zonal wind. (d) The variables include divergence (shading, units: $10^{-5}$ s$^{-1}$) and q (contour, units: $10^{-4}$ kg kg$^{-1}$). White dots indicate that divergence anomalies exceed the 95% confidence level. The vectors represent ω (magnified 100 times) and zonal wind. Panel (c) and (f) are the same as panel (a) and (d) respectively, but for Non-Dust days. Composite anomalies of meridional component of the vertical circulation average over 40–60° N, 90–120° E during CH days: (b) The variables include ω (shading, units: Pa s$^{-1}$) and downward transport of westerly momentum (<0, dashed contour, units: $10^{-3}$ m s$^{-2}$). White dots indicate that ω anomalies exceed the 95% confidence level. The vectors represent ω (magnified 100 times) and **meridional** wind. (e) The variables include divergence (shading, units: $10^{-5}$ s$^{-1}$) and q (contour, units: $10^{-4}$ kg kg$^{-1}$). White dots indicate that divergence anomalies exceed the 95% confidence level. The vectors represent ω (magnified 100 times) and **meridional** wind.

**Reply to Referee 2:**

Dust weather in North China has been studied using $PM_{10}$ concentration observation and ERA5 data. Quantitative contribution of the Mongolian cyclone and the cold high to the dust days were given. A common predictor of the two dust weather types was also identified. **The study will benefit the understanding the synoptic meteorological influence on dust weather in North China,** but still some points need to be further clarified.

Major comments:

1. It should be identified that if there are only two dust weather types of the Mongolian cyclone and the cold high which can influence the dust in North China?

*Reply:*

Thank you for your professional advice. The impact of the Mongolian cyclone on dust weather in North China has been widely revealed. **The composite original SLP for dust days without the Mongolian cyclone shows** the influence of a **cold high** (Fig. S1d). Therefore, the two types of Dust days were respectively **named Mongolian Cyclone (MC) type** and **Cold High (CH) type**. We apologize for any misleading in our previous manuscript. In the **revised version**, we describe the **dust days without Mongolian cyclones** as **influenced by other synoptic systems**, **primarily by cold high**.

Previous studies have shown that **there are more than two dust weather types** affecting North China. In addition to the Mongolian cyclone and cold high, synoptic systems such as a pure cold front can also lead to severe sandstorms in North China (Liu et al., 2004). This study **differs from traditional weather classification studies** by not further categorizing weather types beyond the Mongolian cyclone. The circulation anomalies related to the two types of dust weather are explored **to identify the common predictor**, which provides reference for dust weather forecasting and climate prediction. We have revised the relevant expressions and discussed in Section 6.

*Related References:*

Liu, J., Qian, Z., Jiang, X., and Zheng M.: A Study on Weather Types of Super

Severe Dust Storms in North China, Plateau Meteor, 23, 540–547, 2004 (in Chinese).

*Revision:*

**p. 3, line 62–69:** … more attention (Yin et al., 2023a). Research on subjective and objective classifications of dust weather has been conducted (Liu et al., 2004; Yun et al., 2013; Yi et al., 2021). These studies indicate that besides the Mongolian cyclone, **other synoptic systems such as the cold high** also played significant roles in causing dust weather in North China.

This study used $PM_{10}$ concentrations as indicators to investigate the differences in the intensity of dust weather caused by the **Mongolian cyclone** and **other synoptic systems**…

**p. 4, line 111–122:** The primary surface synoptic system leading to dust weather in NC was the Mongolian cyclone (Li et al., 2022). Additionally, **other synoptic systems** could also contribute to dust weather affecting NC (Liu et al., 2004). By **objectively identifying the presence of the Mongolian cyclone**, Dust days were further classified into two categories. As depicted in the **original SLP fields** for the two types, the **main surface synoptic systems** for the two types of Dust days were the **Mongolian cyclone** and **cold high** respectively (Fig. S1c, d). Therefore, the two types of Dust days were respectively **named Mongolian Cyclone (MC) type** and **Cold High (CH) type**. Dust days caused by Mongolian cyclones (MC type) accounted for a significant portion during the spring seasons from 2015 to 2023, at 61.7%. **Other synoptic systems**, **mainly cold high systems** (CH type), accounted for 38.3% of the total Dust days…

[Figure]

**Figure S1.** (c) **Composites** of **original SLP** (shading, units: hPa) and UV10 (vectors, units: m s$^{-1}$) during MC days. Panel (d) is the same as panel (c) but for CH days. The green boxes in panel (a)–(d) represent NC.

**p. 13, line 291:** This study **differs from traditional weather classification studies** by **not further categorizing weather types beyond the Mongolian cyclone.** The circulation anomalies related to the two types of dust weather were explored to identify the common predictor…

2. It's not so clear that how to get daily maximum $PM_{10}$ concentration in North China from hourly observation data at the stations? If only maximum value in one station is considered, maybe the regional characteristics can not be represented due to local effect. *Reply:*

(1) The method for selecting the daily maximum $PM_{10}$ concentration in this study is as follows: First, **the daily maximum value for each station** in North China is selected from the hourly observed $PM_{10}$ concentration data. Then, **the station with the highest $PM_{10}$ concentration** in North China is chosen, and the maximum value at that station is used as a reference for selecting dust day.

(2) The **regional mean values of the daily maximum $PM_{10}$ concentrations in North China** have been calculated. The daily regional average values indicate that the MC-type dust days still show relatively higher average $PM_{10}$ concentrations, medians, and outliers compared to the CH-type dust days. **The results obtained do not show significant differences** compared to using the maximum $PM_{10}$ concentration as the dust indicator in one station. The **actual hourly observed values of $PM_{10}$ concentrations** are retained as the discriminant indicator. Using the actual $PM_{10}$ concentration observations from the station with the highest value can **clearly demonstrate the drastic increase** in $PM_{10}$ concentrations caused by dust weather, thereby **better identifying dust day**. **In previous studies**, the **thresholds for $PM_{10}$ concentration** in dust weather were also defined **based on hourly actual values** (Wan et al., 2004; Wang et al., 2008). **Additionally**, through **composite analysis** of **maximum $PM_{10}$ concentrations** during MC and CH days, the results show that **most parts of North China exhibit high $PM_{10}$ concentrations (Fig. S3)**. The composite results of dust days show that North China **still exhibits regional characteristics** of dust weather.

We **added the explanation** of the method used to obtain the daily maximum $PM_{10}$ concentration in North China.

*Related References:*

Wan, B., Kang, X., Zhang, J., Tong, Y., Tang, G., and Li, X.: Research on classification of dust and sand storm basic on particular concentration, Environ. Monit. China, 20, 8–11, https://doi.org/10.3969/j.issn.1002-6002.2004.03.003, 2004 (in Chinese).

Wang, Y. Q., Zhang, X. Y., Gong, S. L., Zhou, C. H., Hu, X. Q., Liu, H. L., Niu, T., and Yang, Y. Q.: Surface observation of sand and dust storm in East Asia and its application in CUACE/Dust, Atmos. Chem. Phys., 8, 545–553, https://doi.org/10.5194/acp-8-545-2008, 2008.

*Revision:*

**p. 3, line 90:** During dust events, $PM_{10}$ concentrations significantly increase, at least doubling or even increasing by tens of times (Dulam et al., 2014). In this study, **the maximum $PM_{10}$ concentration** was utilized to **confirm the occurrence of dust weather in NC**. In order to better identify dust days, the daily maximum $PM_{10}$ concentrations were selected from **the actual hourly observed values** for each station in NC. **The station with the highest $PM_{10}$ concentration in NC** was chosen, and the **maximum value at that station** was used as a reference for selecting dust day…

[Figure]

**Figure S3.** (a) Composite distribution of observed daily maximum $PM_{10}$ concentrations (scatter, unit: $\mu g\ m^{-3}$) during MC days. Panel (b) is the same as (a) but for CH days. The shading in panel (a) indicates NDVI in March 2023. (c) Composite distribution of observed daily maximum $PM_{10}$ concentrations anomalies (scatter, unit: $\mu g\ m^{-3}$) during MC days. Panel (d) is the same as (a) but for CH days. The green boxes in panel (a)–(d) represent NC.

3. In many previous studies, dust weather phenomenon and visibility observation were used to characterize the dust spatial-temporal distribution, which may lead different result with this study only using $PM_{10}$ concentration. For example, April is the most frequent dust weather month in previous studies, but it's May in this manuscript. It should be pointed out and discussed.

*Reply:*

Thank you for your suggestion. The **definition of dust days using $PM_{10}$ concentration** has been **applied in previous studies** (Krasnov et al., 2016; Jenkins et al., 2017). However, there is currently **no unified criteria** for this. Traditional dust weather defined by visibility and other weather phenomena **may differ** from dust weather identified solely by $PM_{10}$ concentration. Additionally, $PM_{10}$ concentration has been more widely observed since 2015, so the analysis in this paper covers the period from 2015 to 2023. The **increase in dust days in May** may be a result of **recent years**. These conclusions indeed require further research and discussion. **Figure 1** has been **revised**, and the differences in results have been **pointed out and discussed** in Section 6.

*Related References:*

Jenkins, G. S. and Diokhane, A. M.: WRF prediction of two winter season Saharan dust events using $PM_{10}$ concentrations: Boundary versus initial conditions, Atmos. Environ., 167, 129-142, https://doi.org/10.1016/j.atmosenv.2017.08.010, 2017.

Krasnov, H., Katra, I., and Friger, M.: Increase in dust storm related PM10 concentrations: A time series analysis of 2001–2015, Environ. Pollut., 213, 36-42, https://doi.org/10.1016/j.envpol.2015.10.021, 2016.

*Revision:*

**p. 13, line 291:** … Furthermore, dust weather **identified by $PM_{10}$ concentrations may differ from** traditional dust weather **defined based on visibility and other meteorological phenomena.** For example, in the years 2015–2023, Dust days defined by $PM_{10}$ concentrations were **most frequent in May, rather than in April as seen in previous studies.** However, the increase in the number of dust days in May may be a recent trend that requires further study.

[Figure]

**Figure 1.** (a) Boxplots of daily maximum $PM_{10}$ concentrations (units: µg m$^{-3}$) in NC during MC days (pink) and CH days (orange). The cyan dashed lines and blue dots in the boxplot represent average $PM_{10}$ concentrations and outlier values. Density distributions of $PM_{10}$ concentrations are shown by pink and orange shadings for MC days and CH days respectively. (b) The composite differences of observed daily maximum $PM_{10}$ concentrations (scatter, units: µg m$^{-3}$) during MC days relative to CH days. The green box indicates NC. (c) The temporal distribution of MC days, CH days and Non-Dust days in spring from 2015 to 2023.

4. There is strict definition of dust weathers in meteorological society with several grads of floating dust, blowing dust, sand and dust storm, severe sand and dust storm. Its well know that high $PM_{10}$ concentration is a major result from dust weather, but the threshold value need to be investigated further to match with the meteorological definition of dust weather. Also, $PM_{2.5}$ and $PM_{10}$ ratio should considered to remove anthropogenic aerosol impact.

*Reply:*

(1) Previous studies have investigated the relationship between **$PM_{10}$ concentration thresholds and traditional dust weather levels**, but a **unified standard has not yet been established**. For hourly $PM_{10}$ concentrations, one set of criteria is as follows: Suspended dust: 200 µg m$^{-3}$ ≤ $PM_{10}$ (with very low wind speed); **Blowing dust**: **200 µg m$^{-3}$** ≤ $PM_{10}$ < 5500 µg m$^{-3}$; Sand and dust storm: 5500 µg m$^{-3}$ ≤ $PM_{10}$ < 15000 µg m$^{-3}$; Severe sand and dust storm: 15000 µg m$^{-3}$ ≤ $PM_{10}$ (Wang et al., 2008). Another standard is as follows: Suspended dust: 600 µg m$^{-3}$ ≤ $PM_{10}$ < 1000 µg m$^{-3}$; **Blowing dust**: **1000 µg m$^{-3}$** ≤ $PM_{10}$ < 2000 µg m$^{-3}$; Sand and dust storm: 2000 µg m$^{-3}$ ≤ $PM_{10}$ < 4000 µg m$^{-3}$; Severe sand and dust storm: 4000 µg m$^{-3}$ ≤ $PM_{10}$ (Wan et al., 2004).

In our study, we determined two $PM_{10}$ concentration thresholds, 500 μg m$^{-3}$ and 1000 μg m$^{-3}$, by calculating the first and third quartiles of the daily maximum $PM_{10}$ concentrations from 2015 to 2023. According to the variation of daily maximum $PM_{10}$ concentrations, days with peak $PM_{10}$ concentrations exceeding 1000 μg m$^{-3}$ were selected as representatives of dust days, while days with minimum $PM_{10}$ concentrations below 500 μg m$^{-3}$ were chosen as representatives of non-dust days. **Comparing our results with previous research**, **dust days** identified by the **1000 μg m$^{-3}$ $PM_{10}$** concentration threshold **primarily correspond to** the traditional meteorological classifications of **blowing dust, sand and dust storm,** and **severe sand and dust storm**. Further studies are needed regarding the **matching** between the **$PM_{10}$ concentration thresholds** and the **traditional dust weather levels**. This has been **discussed** in Section 6.

(2) Considering the potential **influence of anthropogenic factors**, we obtained the **$PM_{2.5}$ concentrations at the same site and time as the maximum $PM_{10}$ concentrations** on the Dust days, and calculated the **$PM_{2.5}/PM_{10}$ ratio**. In North China, the average $PM_{2.5}/PM_{10}$ ratio typically ranges from 0.5 to 0.7 and exhibits distinct seasonal variations, being lower in spring (Wang et al., 2015). Among the dust days we examined, **only two days in 117 days had a $PM_{2.5}/PM_{10}$ ratio exceeding 0.5** (Fig. R1). **98.3%** of Dust days are consistent with the result in our study (Fig. R1). To eliminate the influence of anthropogenic aerosols, we have **excluded these two days**. Subsequently, we **updated all relevant figures and data**, with the **overall findings remaining largely consistent with the original results**. The method of using the $PM_{2.5}$ and $PM_{10}$ ratio to remove anthropogenic aerosol impact has been **added to Section 2**.

[Figure]

**Figure R1.** The temporal distribution of MC days, CH days and Non-Dust days in spring from 2015 to 2023. The **red square** indicates the Dust day with **$PM_{2.5}/PM_{10}$ ratio exceeding 0.5**, which is **excluded** from the analysis.

*Related References:*

Wan, B., Kang, X., Zhang, J., Tong, Y., Tang, G., and Li, X.: Research on classification of dust and sand storm basic on particular concentration, Environ. Monit. China, 20, 8–11, https://doi.org/10.3969/j.issn.1002-6002.2004.03.003, 2004 (in Chinese).

Wang, Y. Q., Zhang, X. Y., Gong, S. L., Zhou, C. H., Hu, X. Q., Liu, H. L., Niu, T., and Yang, Y. Q.: Surface observation of sand and dust storm in East Asia and its application in CUACE/Dust, Atmos. Chem. Phys., 8, 545–553, https://doi.org/10.5194/acp-8-545-2008, 2008.

Wang, Y. Q., Zhang, X. Y., Sun, J. Y., Zhang, X. C., Che, H. Z., and Li, Y.: Spatial and temporal variations of the concentrations of $PM_{10}$, $PM_{2.5}$ and $PM_1$ in China, Atmos. Chem. Phys., 15, 13585–13598, https://doi.org/10.5194/acp-15-13585-2015, 2015.

*Revision:*

**p. 3, line 90:** … Considering potential **anthropogenic influences**, the **$PM_{2.5}$ concentrations** were obtained **at the same site and time as the maximum $PM_{10}$ concentrations** on the Dust days, and the **$PM_{2.5}$/$PM_{10}$ ratios** were calculated…   In NC, the average $PM_{2.5}$/$PM_{10}$ ratio typically ranges from 0.5 to 0.7 and exhibits distinct seasonal variations, being lower in spring (Wang et al., 2015). To **eliminate the influence of anthropogenic aerosols**, **two days were excluded** from the selected Dust days as their **$PM_{2.5}$/$PM_{10}$ ratio exceeded 0.5**.

**p. 13, line 291:** … Previous studies have investigated the relationship between $PM_{10}$ concentration thresholds and traditional dust weather levels, but a **unified standard has not yet been established**. According to the standards used in previous studies (Wan et al. 2004; Wang et al., 2008), Dust days identified by the **1000 μg m$^{-3}$** $PM_{10}$ concentration threshold primarily **correspond to** the traditional meteorological classifications of **blowing dust**, **sand and dust storm**, and **severe sand and dust storm**…

5. How about the weather with the maximum $PM_{10}$ concentration between 500 and 1000 μg/m$^3$?
*Reply:*

The maximum $PM_{10}$ concentrations **between 500 and 1000 μg/m³** represents a **transitional phase** during the **development** and **cessation** of **dust weather**. In this study, we only focus on **extreme situations** of dust weather. The **dust days** selected

based on the peak values of $PM_{10}$ exceeding 1000 μg/m³ **represent the days most significantly impacted** by dust in North China **during typical dust events**. Aiming at **capture the most significant anomalous circulation patterns** and **extreme situations**, this study primarily analyzes the anomalous circulation features **during the peak impact** of dust weather. The circulation anomalies during the transitional phase of dust weather in NC are not included. Explanations regarding the weather associated with $PM_{10}$ concentrations between 500 and 1000 μg/m³ have been **added in Section 6**.

*Revision:*

**p. 13, line 291:** … The study designates the **peak $PM_{10}$ concentration** during periods when the daily maximum $PM_{10}$ concentration exceeds 1000 μg m$^{-3}$ as Dust days, **representing a typical dust event**. This aims to capture the most significant circulation anomalies and **extreme conditions** during dust events. The **transitional phase** during the **development** and **cessation** of **dust weather** in NC has **not been included** in this study…

6. The area of concern in this paper is limited at 34-42°N, 105-120°E, but the dust weather is a large-scale process that has an important impact on the entire northern region of China, and the two types of atmospheric circulation such as cold high and Mongolia cyclone will also affect Inner Mongolia and Northeast China. The introduction should also be supplemented by a statement of the rationale for the selection of the region, how it differs from other work, and the geographical importance of the selected area of analysis.

*Reply:*

Thank you for your suggestion. Dust weather significantly impacts the entire northern region of China. In this study, **unlike previous research** that **uniformly studied all dust weather events** in the **entire** northern region of China, we have selected the North China region (34-42°N, 105-120°E) for analysis. This region, **apart from the northwestern region** of China, experienced **the highest frequency and intensity** of dust weather events (Zhang et al., 2023). The **average $PM_{10}$ concentrations** in this area during spring are relatively high, **surpassing those in the northeastern region** of China (Fig. R2). Furthermore, this region is **densely populated**, economically developed, and plays a vital role in China's **economy**, **politics**, **culture**,

and **agriculture**. Therefore, enhancing the understanding and forecasting of dust weather in North China holds significant importance. The reasons for selecting this region, differences from other studies, and the geographical significance of the chosen analysis area have been **added in the introduction** section.

[Figure]

**Figure R2.** Spring mean $PM_{10}$ concentrations in 2021 (cited from Li et al., 2022; Fig. 2e). The **red box** represent **NC** in our study.

*Related References:*

Li, J. D., Hao, X., Liao, H., Yue, X., Li, H., Long, X., and Li, N: Predominant type of dust storms that influences air quality over northern China and future projections, Earth's Future, 10, e2022EF002649, https://doi.org/10.1029/2022EF002649, 2022.

Zhang, X. X., Lei, J. Q., Wu, S. X., Li, S. Y., Liu, L. Y., Wang, Z. F., Huang, S. Y., Guo, Y. H., Wang, Y. D., Tang, X., and Zhou, J.: Spatiotemporal evolution of aeolian dust in China: An insight into the synoptic records of 1984–2020 and nationwide practices to combat desertification, Land. Degrad. Dev., 34, 2005–2023, https://doi.org/10.1002/ldr.4585, 2023.

*Revision:*

**p. 1, line 29:** … adversely affecting traffic safety.

North China (NC; 34–42°N, 105–120°E) is the region, apart from Northwest China, where spring dust weather **is most frequent**, **intense**, and has the **highest average $PM_{10}$ concentration** (Li et al., 2022; Zhang et al., 2023). **In contrast to previous studies** that examined dust weather across the **entire** northern region of China, this study specifically focuses on NC. With a dense population, developed economy, and playing a vital role in China's **economy**, **politics**, **culture**, and **agriculture,** studying dust weather in NC is of great importance. In response to global warming …

Specific comments:

1. In the Data and method section, it is recommended to add a description of the study area and the number of PM$_{10}$ stations used, as well as a distribution map.

*Reply:*

Thank you for your advice. The description of the study area and the number of PM$_{10}$ stations used have been **added** in **Section 2**. The **distribution** of PM$_{10}$ monitoring stations can be seen from **Fig. 1b**.

[Figure]

**Figure 1.** (b) The composite differences of observed daily maximum PM$_{10}$ concentrations (scatter, units: μg m$^{-3}$) during MC days relative to CH days. The green box indicates NC.

*Revision:*

**p. 3, line 80-81:** … The **study area is located in NC**, specifically within the range of **34–42°N, 105–120°E**. PM$_{10}$ and PM$_{2.5}$ concentration data from **556 stations in NC** have been utilized for selecting dust weather days, with negative and missing values excluded. The Normalized Difference Vegetation Index …

2. From table 2, it can be seen that the correlation between I_SAT and I_Gust10 and PM$_{10}$ concentration is significantly higher than that of I_ACA-CA. There is no explanation in the text.

*Reply:*

We are sorry for not making it clear. It is **reasonable** that there is a **stronger correlation** between near-surface meteorological conditions and PM$_{10}$ concentrations. Similar results have been observed in previous research related to atmospheric particulate matter pollution (Zhong et al., 2022). In this study, we **aim to identify a key**

**anomalous circulation system** that can predict both types of dust weather. **The circulation index I_ACA-CA most highly correlated with PM$_{10}$ concentration** was selected as the common predictor (**R = 0.321**). The absolute correlation coefficient is **higher than I_Z500c** (|R| = 0.205) and **I_Z500a** (|R| = 0.167). Additionally, cyclonic and anticyclonic circulation anomalies at 500 hPa during dust days also significantly correlated with anomalous strong surface winds and temperature. After organizing the logic, **Section 5** has been **rewritten** to avoid misunderstandings.

*Related References:*

Zhong, W. G., Yin, Z. C., and Wang, H. J.: The relationship between anticyclonic anomalies in northeastern Asia and severe haze in the Beijing–Tianjin–Hebei region, Atmos. Chem. Phys., 19, 5941–5957, https://doi.org/10.5194/acp-19-5941-2019, 2019.
*Revision:*

**p. 10, line 217-240:** … **In order to comprehensively predict dust weather of the MC and CH types**, we defined a series of meteorological indices **to explore the common anomalous circulation systems influencing these two dust weather types**… The 500 hPa cyclonic anomaly (**CA**) and anticyclonic anomaly (**ACA**) circulation systems were represented by the 500 hPa geopotential height indices **I_Z500c** and **I_Z500a** (Table 1) …

By **considering CA and ACA together**, calculating the difference in Z500 between them and normalizing it, the index I_ACA-CA was defined. **I_ACA-CA** is significantly correlated with the maximum PM$_{10}$ concentration in NC (**R = 0.321**), with an absolute **correlation coefficient higher than I_Z500c (|R| = 0.205) and I_Z500a (|R| = 0.167)**. During MC days, CH days, and Non-Dust days, the composite values of I_ACA-CA also showed significant differences, corresponding to the composite PM$_{10}$ concentration values during these days (Fig. 5a). Furthermore, **I_ACA-CA** exhibited **significant correlations with meteorological conditions** and horizontal circulation influencing dust weather in NC (Table 2), **consistent with the physical mechanisms** described in Section 4. Therefore, CA and ACA are closely related to dust weather in NC and are key anomalous circulation systems. I_ACA-CA is designated as a common predictor for the two types of Dust days in NC…

3. The lines of Figure 6 are light and similar in color, and there is no illustration of the lines in the figure.

*Reply:*

The colors of the lines in Figure 6 have been darkened, new colors have been applied to enhance differentiation, and relevant line illustrations have been added.

*Revision:*

**p. 13, line 286-290:**

[Figure]

**Figure 6.** Daily I_ACA-CA (blue line) and normalized daily maximum $PM_{10}$ concentrations observed in NC (grey line) in March (Mar), April (Apr) and May from 2015 to 2023. The blue dots indicate the Dust days captured when I_ACA-CA>0 and the corresponding I_ACA-CA value. The pink and yellow vertical lines indicate the MC days and CH days captured when I_ACA-CA>0, respectively. The red "x" marks represent the Dust days that I_ACA-CA failed to capture.

**Reply to Referee 3:**

1. Line 11,line 304, what's "3.15" super dust storm?  -> should likely be: a super dust storm occurred in 15 March 2021 (hereafter "3.15" dust for short) ? I would not to use such kind of strange abbreviate.

*Reply:*

Thank you for your suggestion. The abbreviation "3.15" has been revised.

*Revision:*

**p. 1, line 11:** … and the economy. A super dust storm **occurred on 15 March 2021** raised Beijing's $PM_{10}$…

**p. 14, line 304:** … its intensity. For the super dust storm **occurred on 15 March 2021**, although the…

2. Introduction, mentioned some numbers of PM concentration, to avoid misleading you would better to clearly indicate how these numbers obtained, such as of the maximum concentration among station observation or as of the peak value in center of the system, or as of the means averaging over a specific region in north China or other region of interest.

*Reply:*

$PM_{10}$ concentrations in line 32–33 in the introduction refer to **hourly observed $PM_{10}$ concentrations**, cited from the article " Why super sandstorm 2021 in North China? " (Yin et al., 2022). The $PM_{10}$ concentrations in line 60–61 represent the **daily maximum $PM_{10}$ concentrations among station observations in North China** (34–42°N, 105–120°E), which are derived from China National Environmental Monitoring Centre. The corresponding concentrations are shown in Fig. S2. The explanation on how the $PM_{10}$ concentrations were obtained has been added.

*Related References:*

Yin, Z. C., Wan, Y., Zhang, Y. J., and Wang, H. J.: Why super sandstorm 2021 in North China?, Natl. Sci. Rev., 9, nwab165, https://doi.org/10.1093/nsr/nwab165, 2022.

*Revision:*

**p. 2, line 31–33:** … an absence for more than 10 years in NC (Zhang et al., 2022). During 14–16 March 2021, the **hourly observed $PM_{10}$ concentration** exceeded the monitoring threshold in Ulanqab ($>9985\ \mu g\,m^{-3}$) and reached extraordinarily high value

in Beijing (>7400 μg m$^{-3}$; Yin et al., 2022).

**p. 2, line 58–62:** … all instances of dust weather. According to **the daily maximum observed PM$_{10}$ concentration in NC,** a dust event caused by a cold high resulted in a relatively low PM$_{10}$ concentration of 1247 μg m$^{-3}$ on 14 March 2023 (Fig. S1a). This event went largely unnoticed. However, on 22 March 2023, a severe dust storm, brought by a Mongolian cyclone (Fig. S1b), led to higher PM$_{10}$ concentrations of 9993 μg m$^{-3}$, garnering more attention (Yin et al., 2023a).

3. Fig. 1b, is it more reasonable to show composite of cyclones and cold highs separately? Because the Mongolian cyclones and cold highs are not counterparts but independent in physics

*Reply:*

Thank you for your suggestion. The **composite results of PM$_{10}$ concentrations** for MC and CH types are shown **separately** in **Fig. S3**. Figure 1b demonstrates the spatial distribution **differences** of composite PM$_{10}$ concentrations for MC and CH types. This study focuses on **highlighting the differences in dust intensity** caused by the two types. Additionally, the **boxplots** of daily maximum hourly PM$_{10}$ concentrations for the two types have been **separately** displayed in Fig. 1a. The emphasis on the differences in PM$_{10}$ concentrations between the two types has been added to Section 3.

*Revision:*

**p. 4, line 98:** 3 PM$_{10}$ concentration **differences** between regional synoptic systems

**p. 4, line 123–126:** There are **distinct differences in PM$_{10}$ concentrations** between the MC and CH types. **Both** of the two types exhibited **high PM$_{10}$** concentrations in NC (**Fig. S3**). Compared to the CH type, the **MC type** resulted in **higher** PM$_{10}$ concentrations and showed **more pronounced extremes** (Fig. 1a). The outliers in the PM$_{10}$ concentration boxplot for MC type in Fig. 1a included the severe dust storms on March 15 2021, and March 22 2023… From the **spatial distribution differences** in PM$_{10}$ concentrations, it can be observed that the MC type resulted in relatively **higher** PM$_{10}$ concentrations in NC, especially in its northern region (Fig. 1b).

[Figure]

**Figure 1.** (a) Boxplots of daily maximum $PM_{10}$ concentrations (units: μg m$^{-3}$) in NC during MC days (pink) and CH days (orange). The cyan dashed lines and blue dots in the boxplot represent average $PM_{10}$ concentrations and outlier values. Density distributions of $PM_{10}$ concentrations are shown by pink and orange shadings for MC days and CH days respectively. (b) The composite differences of observed daily maximum $PM_{10}$ concentrations (scatter, units: μg m$^{-3}$) during MC days relative to CH days. The green box indicates NC.

[Figure]

**Figure S3.** (a) Composite distribution of observed daily maximum $PM_{10}$ concentrations (scatter, unit: μg m$^{-3}$) during MC days. Panel (b) is the same as (a) but for CH days. The shading in panel (a) indicates NDVI in March 2023. (c) Composite distribution of observed daily maximum $PM_{10}$ concentrations anomalies (scatter, unit: μg m$^{-3}$) during MC days. Panel (d) is the same as (a) but for CH days. The green boxes in panel (a)–(d) represent NC.

4. Section 4, line 143--  , why talk anomalies in circulation relevant to synoptic cyclone and cold high? For example, readers may doubt the reality and reasonablity of the system if you talk typhoon in anomalous SLP fields. It seems better to show composites for original circulation field together with the anomalies, anyway, to avoid the possible confusing between the real synoptic system and the patterns exiting only in anomalous fields.

*Reply:*

We apologize for any misleading in our manuscript. The **identification of the Mongolian cyclone** and the **classification** of the two types of dust days are **based on the original SLP** circulation field, which has been **shown** in **Fig. S1**.

This study focuses on the **atmospheric circulation anomaly patterns** during dust weather occurrences under the influence of two synoptic systems. The anomaly fields provide a **clearer view of the circulation and meteorological conditions** associated with dust weather in North China. The **differences** in circulation between **the two types** can also be **clearly demonstrated** by anomaly fields. The previous wording in the article may lead to misunderstandings, so the revised version **emphasizes the anomalous conditions** of the circulation and **contrasts them with the original circulation fields** for analysis.

*Revision:*

**p. 4, line 114:** As depicted in the **original SLP fields** for the two types, the **main surface synoptic systems** for the two types of Dust days were the **Mongolian cyclone** and **cold high** respectively (Fig. S1c, d).…

**p. 6, line 142:** 4 Large-scale atmospheric circulation **anomalies**

**p. 6, line 143–line 165:** During Dust days of MC and CH types, there were strong **anomalous northerly winds** to the north of NC at the surface (**Fig. 2d, e**). During MC days, **due to the presence of the Mongolian cyclone (Fig. S1c)**, there was a significant **negative SLP anomaly** in the eastern part of East Asia, with a **positive anomaly** in the west (Fig. 2a). The rear part of the **low-pressure anomaly** exhibited strong **anomalous northwest winds** (Fig. 2d). During CH days, the northern part of East Asia exhibited a significant **cold high-pressure anomaly**, with a **low-pressure anomaly** in the south (Fig. 2b). Between the two anomalous circulations, there were **northeasterly anomalous winds** to the north of NC (Fig. 2e). The **easterly wind anomaly**

components **weakened the westerly wind components** of the surface winds in the CH type. For CH days, the **actual wind** direction was **more northerly** compared to MC days (Fig. S1c, d) …

[Figure]

**Figure S1.** (c) **Composites** of **original SLP** (shading, units: hPa) and UV10 (vectors, units: m s$^{-1}$) during MC days. Panel (d) is the same as panel (c) but for CH days. The green boxes in panel (a)–(d) represent NC.

[Figure]

**Figure 2.** (a) **Composite anomalies** of SLP (shading, units: hPa) and SAT (contour, units: K) during MC days. White dots indicate that SLP anomalies exceed the 95% confidence level. Panel (b) and (c) are the same as panel (a) but for CH days and Non-Dust days. (d) Composite anomalies of q (shading, units: 10$^{-3}$ kg kg$^{-1}$) and UV10 (vectors, units: m s$^{-1}$) during MC days. White dots indicate that q anomalies exceed the 95% confidence level. Panel (e) and (f) are the same as panel (d) but for CH days and Non-Dust days. The green boxes in panel (a)–(f) represent NC. S

5. Section 4, need to carefully clarify and refine. Synoptic processes and climate processes are combined/mixed. Case composites, as I see, are essentially synoptic configuration for multi-variables/fields, and can hardly to tell which is the cause and which is the effect.

*Reply:*

Section 4 has been **rewritten** with the following **modifications** based on the suggestions:

(1) We use composite analysis methods aimed at finding the common characteristics of synoptic processes. Section 4 in the previous manuscript analyzes from a synoptic perspective. **The revised manuscript mainly analyzes anomalous circulation conditions**. It explains the distribution of anomalous circulation fields, **their correlation with the original circulation fields**, and elucidates how **anomalous meteorological conditions can provide favorable conditions** for dust weather in North China.

(2) In the case composites, the configuration of multivariable fields is simultaneous, making it difficult to distinguish causality. **The revised Section 4 has primarily described the coexistence of anomalous conditions.** The anomalies in different atmospheric pressure levels are described **separately**. Additionally, the previous statement in Section 4 suggesting that mid-to-high-level troposphere circulation leads to the configuration of synoptic systems and meteorological conditions near the surface has been revised. In the **revised Section 5**, it is demonstrated through the calculation of **correlation coefficients** that there is a **correlation** between **anomalies in mid-to-high-level circulation** and **anomalies in near-surface circulation and meteorological conditions**.

*Revision:*

**p. 6, line 143–line 165:** … Although there were significant differences in the surface circulation anomaly patterns between the MC and CH types, some similar features could be observed in the mid to upper troposphere. The MC type and CH type **displayed intensified westerly winds** over the mid-latitude East Asia region at 200 hPa (Fig. 3a, b). At 500 hPa, both the MC and CH types **exhibited cyclonic anomalies** to the north of NC and anticyclonic anomalies to the east of NC (Fig. 3a, b). Compared to

the CH type, the MC type showed stronger negative **geopotential height anomalies** at 500 hPa, with the center located in the northern part outside of NC (Fig. 3a). In contrast, the geopotential height anomalies to the north of NC at 500 hPa were weaker for the CH type (Fig. 3b). Since this 500 hPa cyclonic anomaly is located adjacent to the northern part of NC and over the sparsely vegetated external dust source area to the northwest of NC (Fig. S3a), we **hypothesize** that this **500 hPa cyclonic anomaly (CA)** is a **key anomalous circulation system** influencing dust activities in NC…

**p. 10, line 230–line 240:** The 500 hPa cyclonic anomaly (**CA**) and anticyclonic anomaly (**ACA**) circulation systems were **represented by** the 500 hPa geopotential height indices I_Z500c and I_Z500a (Table 1). While **I_Z500c** showed significant **correlations with other related meteorological indices**, it couldn't explain the anomalies in VATD and PBLH well on Dust days (Table 2). If only CA is considered, it may not be sufficient to provide the thermodynamic instability conditions for dust weather. When considering the role of **ACA**, it can be observed that **I_Z500a exhibited a stronger correlation with VATD and PBLH**, with correlation coefficients reaching 0.639 and 0.534, respectively (Table 2) … By considering CA and ACA together, calculating the difference in Z500 between them and normalizing it, the index I_ACA-CA was defined… Furthermore, **I_ACA-CA** exhibited **significant correlations** with **meteorological conditions** and **horizontal circulation** influencing dust weather in NC (Table 2), **consistent with** the physical mechanisms described in **Section 4**.

6. Dust are uplifted locally or transported into the region? Omega alone seems cannot explain why there are high PM concentration in both Mongolian cyclone and cold high cases, and the anomalous omega are also of very large scale extending toward the far west arid lands beyond the region of NC. Or simply caused by the horizontal wind speed at surface? Anyway, need to clarify the logics and consistence.
*Reply:*

(1) **This study primarily analyzes the transport of external dust to NC.** Previous studies have shown that **a significant portion** of the dust in NC **originates from external transport**, specifically from Outer Mongolia (Yin et al., 2022; Chen et al., 2023). 500 hPa cyclonic anomaly (**CA**) is located adjacent to the northern part of NC and **over the sparsely vegetated external dust source area** to the northwest of

NC. The analysis in this study confirms that the 500hPa cyclonic anomaly (CA), which is associated with the external dust transport into North China (NC), is the key anomalous circulation system. The **vertical circulation anomalies associated with CA** can provide **favorable conditions** for **dust emission** from the source areas and the **outward dispersion** of dust.

(2) For the **MC type**, the **zonal component** of vertical circulation anomalies is **more significant**. In the case of the **CH type**, the zonal component of vertical circulation anomalies is **weaker**, but the **meridional component** is more **pronounced**. The vegetation in the northwestern dust source areas is sparser compared to the northern regions, which may partially explain why the dust intensity of the MC type is greater than that of the CH type. In the revised version, we have **replaced Fig. 4b** and **Fig. 4e** with a **more significant vertical circulation anomalies** of the **meridional component** during **CH days**.

After **organizing the logic**, the relevant content has been **rewritten**.

*Related References:*

Chen, S. Y., Zhao, D., Huang, J. P., He, J. Q., Chen, Y., Chen, J. Y., Bi, H. R., Lou, G. T., Du, S. K., Zhang, Y., and Yang, F.: Mongolia Contributed More than 42% of the Dust Concentrations in Northern China in March and April 2023, Adv. Atmos. Sci., 40, 1549–1557, https://doi.org/10.1007/s00376-023-3062-1, 2023.

Yin, Z. C., Wan, Y., Zhang, Y. J., and Wang, H. J.: Why super sandstorm 2021 in North China?, Natl. Sci. Rev., 9, nwab165, https://doi.org/10.1093/nsr/nwab165, 2022.

*Revision:*

**p. 6, line 166–line 178:** … Since this 500 hPa cyclonic anomaly is located adjacent to the northern part of NC and over the **sparsely vegetated external dust source area to the northwest of NC** (Fig. S3a), we hypothesize that this 500 hPa cyclonic anomaly (CA) is a key anomalous circulation system influencing dust activities in NC. The vertical circulation structure of CA was further analyzed…

The **vertical circulation anomalies** associated with **CA** were primarily related to **the emission and transport of dust** from **the external dust source areas outside of NC**…

**p. 8, line 191–line 198:** The **east-west contrast** in the vertical structure of the CA

was **weaker** for the **CH** type compared to the MC type (**Fig. S5a**), whereas the **north-south contrast** for the CH type was **more pronounced** (**Fig. 4b, e**). For the CH type, the **enhanced anomalous meridional vertical circulation** increased the importance of the **dust source areas to the north of NC** (Fig. S3a). The **vegetation cover** outside of NC along the **northern direction** was relatively **better** than that **along the northwest direction outside of NC**, indicated by higher NDVI (Fig. S3a). This might lead to **lower PM$_{10}$ concentrations** in NC during CH days compared to MC days…

[Figure]

**Figure S5.** Composite anomalies of zonal component of the vertical circulation average over 40–60°N, 90–120°E during CH days: (a) The variables include ω (shading, units: Pa s$^{-1}$) and downward transport of westerly momentum (<0, dashed contour, units: 10$^{-3}$ m s$^{-2}$). White dots indicate that ω anomalies exceed the 95% confidence level. The vectors represent ω (magnified 100 times) and zonal wind. (b) The variables include divergence (shading, units: 10$^{-5}$ s$^{-1}$) and q (contour, units: 10$^{-4}$ kg kg$^{-1}$). White dots indicate that divergence anomalies exceed the 95% confidence level. The vectors represent ω (magnified 100 times) and zonal wind.

[Figure]

**Figure 4.** Composite anomalies of zonal component of the vertical circulation average over 40–60°N, 90–120°E during MC days: (a) The variables include ω (shading, units: Pa s$^{-1}$) and downward transport of westerly momentum (<0, dashed contour, units: 10$^{-3}$ m s$^{-2}$). White dots indicate that ω anomalies exceed the 95% confidence level. The vectors represent ω (magnified 100 times) and zonal wind. (d) The variables include divergence (shading, units: 10$^{-5}$ s$^{-1}$) and q (contour, units: 10$^{-4}$ kg kg$^{-1}$). White dots indicate that divergence anomalies exceed the 95% confidence level. The vectors represent ω (magnified 100 times) and zonal wind. Panel (c) and (f) are the same as panel (a) and (d) respectively, but for Non-Dust days. Composite anomalies of meridional component of the vertical circulation average over 40–60° N, 90–120° E during CH days: (b) The variables include ω (shading, units: Pa s$^{-1}$) and downward transport of westerly momentum (<0, dashed contour, units: 10$^{-3}$ m s$^{-2}$). White dots indicate that ω anomalies exceed the 95% confidence level. The vectors represent ω (magnified 100 times) and meridional wind. (e) The variables include divergence (shading, units: 10$^{-5}$ s$^{-1}$) and q (contour, units: 10$^{-4}$ kg kg$^{-1}$). White dots indicate that divergence anomalies exceed the 95% confidence level. The vectors represent ω (magnified 100 times) and meridional wind.

7. Fig.2, Z500 and U200, I would like to suggest authors to change legend, say, Z500 shown in contour lines, so helpful to demonstrate mid-troposphere trough and ridge, and helpful to explain near surface cyclone before the trough/behind the ridge.

*Reply:*

Thank you for your suggestion. The **legend** for **Figure 2 (a)–(c)** has been **changed**. In the revised version, **Z500** is shown in **contour lines (Fig. 3)**. The mid-tropospheric trough and ridge are well depicted in the modified version, contrasting with the original circulation field of Z500 **(Fig. R1)**.

[Figure]

**Figure R1.** (a) Composites of **Z500 (contour**, units: geopotential meter, gpm) and U200 (shading, units: m s⁻¹) during MC days. Panel (b) and (c) are the same as panel (a) but for CH days and Non-Dust days. The green boxes in panel (a)–(c) represent NC.

*Revision:*

**p. 7, line 179–line 184:**

[Figure]

**Figure 3.** (a) Composite anomalies of **Z500** (**contour**, units: geopotential meter, gpm) and U200 (shading, units: m s$^{-1}$) during MC days. White dots indicate that U200 anomalies exceed the 95% confidence level. Panel (b) and (c) are the same as panel (a) but for CH days and Non-Dust days. The green boxes in panel (a)–(c) represent NC.

8. Section 5, all predictors are simultaneous fitting variables identified in composite analysis. Readers may find of interest if you demonstrate how the cyclone and cold highs generates/develops/moves/strengthens and construct a set of predictor and predicting the dust in advance accordingly.

*Reply:*

In Section 5, based on simultaneous variables and correlation analysis, we found that the 500hPa cyclonic anomaly (**CA**) and anticyclonic circulation anomaly (**ACA**) are **important anomalous circulation systems** influencing the two types of dust weather in North China. Therefore, we propose **a common predictor**, I_ACA-CA, for the two types of dust weather. I_ACA-CA provide a reference for **seasonal prediction** of dust weather. The ability of the **C3S seasonal forecast model** to reproduce I_ACA-CA was further assessed. The I_ACA-CA calculated by **ECMWF, DWD, and MF** seasonal forecast models **with a one-month lead captured 46.1%, 52.2%, and 51.3% of spring dust days** when positive.

**In addition, I_ACA-CA shows a dust signal two days in advance of dust days**, capturing **56.5%, 69.6%,** and **76.5%** of dust days two days in advance, one day in advance, and on the dust day, respectively. **The evolution of CA, ACA and related atmospheric circulation anomalies before** MC days and CH days was **further investigated** (**Fig. 7**). **The movement and development of the Mongolian cyclone and the cold high before** the two types of dust days are also **depicted through the original circulation field** (**Fig. S7**). The development and movement of CA and ACA aligned with the occurrence and development of MC days and CH days in NC. The common predictor I_ACA-CA served as a meaningful indicator for predicting dust weather in NC.

Based on the above, we have **reorganized the logic** of Section 5 and **rewritten** it.

*Revision:*

**p. 10, line 217–line 267:**

… In order to **comprehensively predict dust weather** of the MC and CH types, we **defined a series of meteorological indices** to **explore the common anomalous circulation systems** influencing these two dust weather types… **I_ACA-CA is designated as a common predictor for the two types of Dust days in NC**… In

correspondence with the positive I_ACA-CA observed **two days, one day (I_ACA-CA>its one standard deviation), and zero day (I_ACA-CA>its one standard deviation) in advance**, successful **capture rates of 56.5%, 69.6%, and 76.5% for Dust days** were achieved. These high percentages suggest that the reinforced positive I_ACA-CA significantly contributed to the high $PM_{10}$ concentrations in NC.

The **evolution** of **CA, ACA** and related atmospheric circulation anomalies **before MC days and CH days** was further investigated (Fig. 7). Prior to both types of Dust days, CA and ACA **moved** eastward towards NC (Fig. 7). For MC type, CA and ACA from western Siberia and Lake Baikal gradually **strengthened** as they moved eastward (**Fig. 7a–d**). **The development of the Mongolian cyclone** intensified (**Fig. S7a–d**), accompanied by an eastward strengthening of the associated surface low-pressure anomaly and cyclonic winds to the northwest of NC (Fig. 7a–d). As the surface low pressure anomaly moved eastward, the anomalous southerly winds north of NC shifted to anomalous northerly winds (Fig. 7c, d). The **actual wind** directions **changed** from westerly to northwesterly (Fig. S7c, d). For CH type, CA and ACA were relatively positioned more to the east (Fig. 7e), with CA moving eastward from central Siberia and gradually weakening, while ACA moving eastward from Northeast Asia gradually strengthened (Fig. 7e–h). **The cold high intensified as it moved eastward** (Fig. S7e–g), accompanied by an eastward strengthening of the surface high-pressure anomaly (Fig. 7e–g). **One day before** the CH-type dust day, the surface high-pressure anomaly **replaced** the low-pressure anomaly to the north of NC (Fig. 7g, h), and the wind anomalies north of NC **shifted** from southwest to northeast (Fig. 7f–h). The actual wind directions **changed** from westerly to northerly directions (Fig. S7f–h). In summary, the development and movement of CA and ACA aligned with the occurrence and development of MC days and CH days in NC. The common predictor I_ACA-CA served as a meaningful indicator for predicting dust weather in NC.

**p. 13, line 293–line 295:** … The I_ACA-CA calculated by **ECMWF, DWD, and MF seasonal forecast models** with a one-month lead captured around 50% of spring dust days when positive.

[Figure]

**Figure 7.** (a)–(d) Lead composite evolution of Z500 (contour, unit: gpm) anomalies, SLP (shading, unit: hPa) anomalies, and UV850 (vectors, units: m s$^{-1}$) anomalies during MC days. Panel (e)–(h) are the same as panel (a)−(d) but for CH days. The green boxes in panel (a)–(h) represent NC, while the black boxes represent the region for calculating I_ACA-CA.

[Figure]

**Figure S7.** (a)–(d) Lead composite evolution of original Z500 (contour, unit: gpm), SLP (shading, unit: hPa), and UV850 (vectors, units: m s$^{-1}$) during MC days. Panel (e)–(h) are the same as panel (a)–(d) but for CH days. The green boxes in panel (a)–(h) represent NC, while the black boxes represent the region for calculating I_ACA-CA.